# DNA replication licensing factor Cdc6 and Plk4 kinase antagonistically regulate centrosome duplication via Sas-6

Xiaowei Xu[1,*], Shijiao Huang[1,*], Boyan Zhang[1], Fan Huang[1], Wangfei Chi[1], Jingyan Fu[1], Gang Wang[1], Si Li[1], Qing Jiang[1] & Chuanmao Zhang[1]

Centrosome number is tightly controlled during the cell cycle to ensure proper spindle assembly and cell division. However, the underlying mechanism that controls centrosome number remains largely unclear. We show herein that the DNA replication licensing factor Cdc6 is recruited to the proximal side of the centrioles via cyclin A to negatively regulate centrosome duplication by binding and inhibiting the cartwheel protein Sas-6 from forming a stable complex with another centriole duplication core protein, STIL. We further demonstrate that Cdc6 colocalizes with Plk4 at the centrosome, and interacts with Plk4 during S phase. Plk4 disrupts the interaction between Sas-6 and Cdc6, and suppresses the inhibitory role of Cdc6 on Sas-6 by phosphorylating Cdc6. Overexpressing wild-type Cdc6 or Plk4-unphosphorylatable Cdc6 mutant 2A reduces centrosome over-duplication caused by Plk4 overexpression or hydroxyurea treatment. Taken together, our data demonstrate that Cdc6 and Plk4 antagonistically control proper centrosome duplication during the cell cycle.

[1] The MOE Key Laboratory of Cell Proliferation and Differentiation and the State Key Laboratory of Membrane Biology, College of Life Sciences, Peking University, Beijing 100871, China. * These authors contributed equally to this work. Correspondence and requests for materials should be addressed to C.Z. (email: zhangcm@pku.edu.cn).

The centrosome duplicates once per cell cycle to ensure proper chromosome separation during cell division. A mature centrosome consists of a pair of centrioles, and the surrounding pericentriolar material that is comprised of several proteins such as the γ-tubulin ring complex[1]. Centrosome duplication cycle consists of three sequential steps: centriole disengagement in which the paired centrioles lose their orthogonal configuration during mitotic exit and the early G1 phase; centriole duplication and elongation in which the procentriole is synthesized and elongated adjacent to each preexisting parental centriole during S and G2 phases; and centrosome maturation and separation during the G2/M transition, which yields two mature polar centrosomes[2]. Thus, centrosome duplication must be synchronized with other cell cycle events, including DNA replication.

G1-S phase cyclin-dependent kinases (CDKs) CDK2-cyclin E and CDK2-cyclin A, the master kinases that control DNA replication initiation, are also required for the activation of centrosome duplication[3–5], linking centrosome duplication and DNA replication. However, the role of CDK2 in centrosome duplication is not completely understood. Interestingly, several DNA replication initiation proteins that interact with cyclin E and cyclin A are directly involved in centrosome duplication. DNA replication initiation requires sequential recruitment of the pre-replication complex (pre-RC) components ORCs, Cdc6, Cdt1 and the Mcm2–7 complex to the replication sites to licence DNA replication, which ensures one round of DNA replication per cell cycle[6,7]. ORC1 prevents over-duplication of the centrosome by controlling the cyclin E level and cyclin E-dependent centriole re-duplication[8]. MCM5 is recruited to the centrosome by interacting with both cyclin E and cyclin A, and represses centrosome amplification in the S phase-arrested CHO cells[9,10]. Geminin, an inhibitor of DNA replication initiation, prevents centrosome over-duplication in the S phase-arrested human breast cancer cell line MDA-MB-231 (ref. 11). However, it is not clear how the DNA replication initiation regulators participate in centrosome duplication. Furthermore, the relationship between the regulators of DNA replication initiation and the key regulators of centriole biogenesis and centrosome duplication is unknown.

Previous work has revealed a conserved pathway for centriole biogenesis in Caenorhabditis elegans, Drosophila melanogaster and the human. SPD-2 (Cep192 in human) is required to recruit ZYG-1 (Plk4 in human) in C. elegans, Asl (Cep 152 in human) is required to recruit Sak (Plk4 in human) in Drosophila, and Cep192 and Cep152 are both required to recruit Plk4 in human[12]. ZYG-1/Sak/Plk4 sequentially recruits the cartwheel components SAS-6 and SAS-5/Ana2/STIL for centriole assembly, and finally, SAS-4/CPAP for centriole elongation[13–16]. Plk4 kinase triggers the onset of centriole biogenesis[13,17,18]. The Plk4 protein level peaks at mitosis and decreases during G1 phase[19]. The overexpression of Plk4 results in centriole amplification, while its depletion reduces centriole number[13]. Therefore, tight control of the Plk4 level, as well as its activity, is critical for the control of centrosome number[20–23]. Plk4 protein level is strictly controlled by SCF/SLimb ubiquitin ligase mediated degradation to block centriole reduplication[24,25]. Centrosome duplication is initiated from a procentriole formation site known as the cartwheel, which contains nine spokes emanating from a central hub. Sas-6 forms the primary backbone of the cartwheel[26]. The recruitment of Sas-6 to the cartwheel requires Plk4 kinase activity in Drosophila and the human[27,28]. Plk4 phosphorylates STIL to facilitate the recruitment of Sas-6 to the cartwheel in cells from both human and Drosophila[15,29,30]. On the other hand, Plk4 phosphorylates the F-box protein Fbxw5 and stabilizes the Sas-6 protein level by inhibiting SCF-Fbxw5 E3 ligase-dependent Sas-6 ubiquitination and degradation, thus triggering centrosome duplication[31].

Several other Plk4 substrates, including the centrosome protein Cep152 (ref. 32) and the γ-tubulin complex protein GCP6 (ref. 33) are also important for centrosome duplication. However, none of the DNA replication-related proteins have been identified as Plk4 kinase substrates.

We herein identified Cdc6 as a novel Plk4 substrate. Cdc6, one of the pre-RC components, initiates DNA replication by recruiting Mcm2-7 helicase to the DNA replication origin dependent on its ATPase activity[34]. After the initiation of DNA replication, Cdc6 is removed from the replication origin and translocated to the cytoplasm during S phase in a CDK2-phosphorylation-dependent manner[35]. Previous reports show that Cdc6 is detected on centrosomes during S and G2 phases[36,37]. However, the function of Cdc6 in centrosome dynamics remains unknown. In this study, we show that Cdc6 is recruited to the proximal side of the centrioles by cyclin A, and prevents centrosome over-duplication by inhibiting the interaction between Sas-6 and STIL. We demonstrate that Cdc6 colocalizes with Plk4 at the centrosome, and interacts with Plk4 during S phase. Plk4 phosphorylates Cdc6 and suppresses the inhibitory role of Cdc6 in centrosome duplication by releasing Sas-6 from Cdc6, facilitating the interaction between Sas-6 and STIL, thus initiating centrosome duplication. Taken together, our data reveal a novel mechanism in which Cdc6 and Plk4 antagonistically control the centrosome number during the cell cycle by regulating the function of the core centrosome duplication regulator Sas-6.

## Results

**Cyclin A mediates the localization of Cdc6 on centrioles.** Cdc6 has been recently reported to localize to the S and G2 phase centrosomes[37]. By immunofluorescence labelling, we confirmed the centrosomal localization of Cdc6 during S and G2 phases (Fig. 1a; Supplementary Fig. 1a–c). By co-immunostaining Cdc6 with CP110 and Cep97, two centriole distal side-localized proteins[38,39], we found that Cdc6 localized to the proximal side of the centrioles (Supplementary Fig. 1d). Cdc6 existed as two dots on the proximal side of both the disengaged parental centrioles during S and G2 phases (Fig. 1a). We observed that exogenous GFP-tagged Cdc6 also localized to the centrioles similar to endogenous Cdc6 (Fig. 1b). To investigate whether the ATPase activity of Cdc6 is required for its centriolar localization, we mutated lysine at residue 208 to glutamate in its Walker A domain (K208E, denoted WA mutant) to abolish its ATP-binding ability, and mutated the glutamine at residue 285 to a glycine in its Walker B domain (E285G, denoted WB mutant) to abolish its ATP hydrolyzing ability[34]. We found that the WA mutant failed to localize to the centrioles. By contrast, the WB mutant localized to the centrioles similar to WT Cdc6 (Fig. 1b; Supplementary Fig. 1e; Supplementary Movies 1–3). To investigate the cell cycle dependent centrosome localization of Cdc6, we performed a live imaging of cells expressing GFP-Cdc6 from G2 phase to the next G1 phase. The result showed that Cdc6 localized in the nucleus but not the centrioles in G1 phase cells (Supplementary Fig. 1f; Supplementary Movie 4). The fact that Cdc6 localizes to the centriole only in S and G2 phase, but not in G1 phase, suggests that the centrosome-localized Cdc6 functions only in S and G2 phases. Taken together, these results demonstrate that Cdc6 localizes on the proximal side of centrioles depending on its ATP binding domain in S and G2 phases.

Next, we investigated how Cdc6 localizes to the centrosome. It has been reported that cyclin A recruits Orc1 and MCM5 to the centrosome[10]. As cyclin A interacts with Cdc6 and promotes the translocation of Cdc6 from the nucleus to the cytoplasm[35], we asked whether cyclin A also regulates the recruitment of Cdc6 to

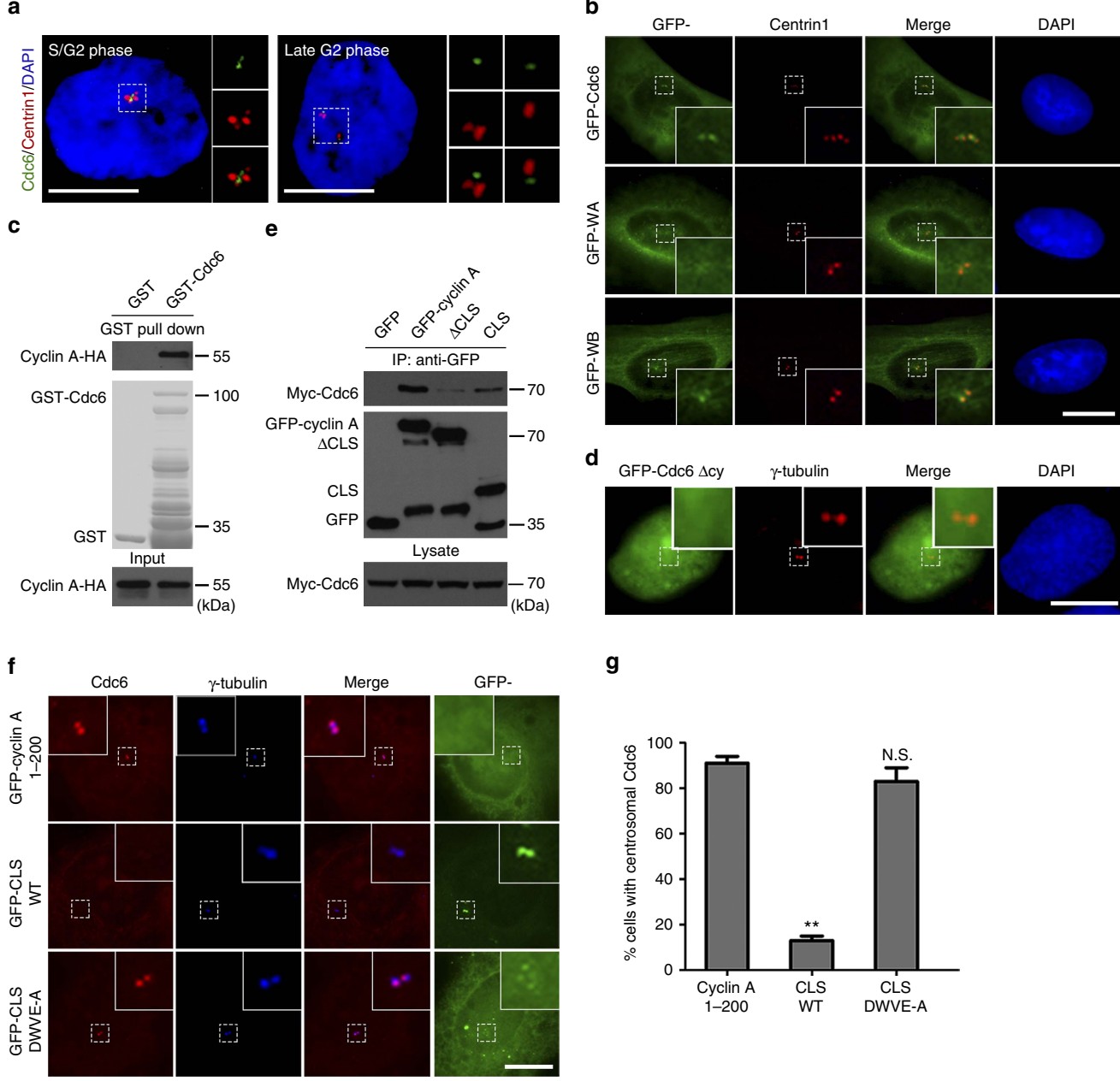

**Figure 1 | Cyclin A mediates the localization of Cdc6 on the proximal side of the centrioles.** (**a**) Cdc6 localizes on the proximal side of centrioles during S and G2 phases. Immunofluorescence showing localization of endogenous Cdc6 (green) and centrin1 (red) in U2OS cells. We defined cells with four contiguous centrin-positive dots as S/G2 phase cells, and cells with two pairs of very distant centrin-positive dots as late G2 phase cells. (**b**) Cdc6 or WB mutant, but not WA mutant, localizes on centrioles. U2OS cells were transfected with GFP-tagged Cdc6 WT, WA or WB mutant and stained with a centrin1 antibody. (**c**) Cdc6 directly binds cyclin A *in vitro*. Sepharose beads coupled with purified GST or GST-tagged Cdc6 were incubated with *in vitro* transcribed and translated HA-tagged cyclin A and analysed by western blotting using a HA antibody. The loading of GST and GST-tagged Cdc6 proteins are shown by Coomassie blue staining. (**d**) Cytoplasm and centrosomal localization of Cdc6 requires the interaction between Cdc6 and cyclin A. U2OS cells were transfected with GFP-tagged Cdc6 Δcy and stained with a γ-tubulin antibody. (**e**) Interactions between Cdc6 and centrosomal localization signal (CLS) domain, or CLS deletion mutant (ΔCLS) of cyclin A are reduced compared to WT cyclin A. HEK293 cells were co-transfected with Myc-tagged Cdc6 and GFP-tagged cyclin A WT, ΔCLS or CLS truncate. The total cell extract was used for immunoprecipitation with a GFP antibody and western blotting with Myc and GFP antibodies. (**f**) The centrosomal localization of cyclin A is required for the centrosomal recruitment of Cdc6. U2OS cells expressing GFP-tagged truncates of cyclin A (amino acids 1-200, CLS or CLS mutant DWVE-A) were stained with Cdc6 and γ-tubulin antibodies. (**g**) Quantitation of cells with centrosomal Cdc6 in **f**. Approximately 300 cells were counted per sample, and three independent experiments were conducted. Data are presented as means ± s.d. **$P < 0.01$; N.S., no significant difference (Student's *t*-test). DNA was stained with DAPI. Scale bars, 10 μm. Insets in **a,b,d** and **f** are high-magnification views of the regions indicated in the low-magnification images.

the centrosome. First, we verified the previously reported interaction between Cdc6 and cyclin A *in vitro* by protein–protein interaction assay (Fig. 1c) and *in vivo* by co-immunoprecipitation (co-IP) (Supplementary Fig. 1g)[35]. We then determined the localization of a Cdc6 mutant (denoted as Cdc6 Δcy) in which the cyclin A binding domain (amino acids 93–100) is deleted, abolishing the interaction between Cdc6 and cyclin A[35]. Cdc6 Δcy was reported to be restricted in the nucleus

and cannot be exported to the cytoplasm, but DNA replication is not affected in Cdc6 Δcy transfected cells[35]. In this case, we tested whether Cdc6 Δcy is defective in the centrosome localization. We observed that Cdc6 Δcy mutant failed to translocate to the cytoplasm, as reported and was not able to localize to the centrosome either (Fig. 1d). To test the possibility that whether Cdc6 recruits cyclin A to centrosome, we depleted Cdc6 by siRNA transfection and found that the centrosome localization of cyclin A was not affected (Supplementary Fig. 1h). These results indicate that the localization of Cdc6 to centrosome requires the interaction between Cdc6 and cyclin A.

To further determine whether Cdc6 is recruited to the centrosome by cyclin A, we set out to observe the centrosome localization of Cdc6 when endogenous cyclin A is absent at the centrosome. It has been reported that overexpression of the centrosomal localization signal (CLS, amino acids 201–255 in cyclin A) fragment of cyclin A can replace endogenous cyclin A at the centrosome[40]. We first compared the interaction between Cdc6 with cyclin A WT, CLS or CLS deletion mutant (denoted herein as ΔCLS)[40]. We found that the interaction between ΔCLS with Cdc6 was dramatically reduced compared to full length WT cyclin A (Fig. 1e), indicating that the interaction between Cdc6 and cyclin A requires CLS domain in cyclin A. However, the CLS domain only had a much weaker interaction with Cdc6 compared to WT cyclin A, indicating that there might be another domain that is required for cyclin A binding to Cdc6 (Fig. 1e). To abolish the centrosomal localization of endogenous cyclin A, we overexpressed GFP-tagged cyclin A CLS fragment (GFP-CLS), and observed that this fragment localized to the centrosome and abolished the endogenous cyclin A centrosomal localization as previous reported[40] (Supplementary Fig. 1i). We also overexpressed GFP-tagged CLS mutant DWVE-A, in which the amino acids DWVE were mutated to AAAA, and found that this mutant failed to localize to the centrosome and did not affect the centrosome localization of endogenous cyclin A (Supplementary Fig. 1i). We then observed the centrosome localization of Cdc6 in GFP-CLS, or GFP-CLS DWVE-A mutant expressing cells. GFP-tagged cyclin A 1–200 fragment was expressed as a negative control (Fig. 1f). Interestingly, we found that, unlike the expression of cyclin A CLS DWVE-A mutant or cyclin A 1–200 fragment, the expression of cyclin A CLS in cells resulted in the absence of Cdc6 at the centrosome (Fig. 1f,g). This result suggests that the substantial loss of endogenous cyclin A at the centrosomes after cyclin A CLS overexpression causes the dislocation of Cdc6 from the centrosomes. Taken together, we conclude that cyclin A interacts with Cdc6 and mediates the localization of Cdc6 at the centrosome.

**Cdc6 negatively regulates centrosome duplication.** To investigate the function of Cdc6 in centrosome, we depleted endogenous Cdc6 by RNA interference with two different siRNAs (Supplementary Fig. 2a). Interestingly, we found that both Cdc6 siRNAs promoted centrosome over-duplication (more than two centrosomes per cell revealed by γ-tubulin staining) and centriole amplification (more than four centrioles per cell revealed by centrin1 staining) (Fig. 2a,b; Supplementary Fig. 2b). We further found that Cdc6 depletion resulted in G2/M phase arrest, indicating that the Cdc6 depletion induced centrosome over-duplication was not the indirect effect of G1/S phase arrest (Supplementary Fig. 2c,d). To investigate the cause of Cdc6 depletion-induced centriole amplification, we stained mother centriole with Cep164 antibody and procentriole with Sas-6 antibody in Cdc6-depleted cells. We found that 24.9% of cells exhibited abnormally amplified Sas-6-positive dots surrounding one Cep164-positive dot (Supplementary Fig. 2e,f), suggesting

that the Cdc6 depletion induced centriole amplification is due to multiple procentriole formation. Accordingly, we observed multipolar mitotic spindles in 22.3% of Cdc6-depleted mitotic cells, which probably resulted from the amplified centrioles caused by Cdc6 depletion (Supplementary Fig. 3a,b). Taken together, these results indicate that Cdc6 prevents centriole amplification.

To further confirm the inhibitory role of Cdc6 in centrosome duplication, we treated cells with hydroxyurea (HU), which causes S phase lengthening and multiple rounds of centrosome duplication during the prolonged S phase[5]. By overexpressing GFP-tagged Cdc6 in the presence of HU, we observed that the percentage of cells with centrosome over-duplication by HU treatment was significantly reduced from 33.9 to 8.5% by Cdc6 WT overexpression (Student's t-test), however, both WA and WB mutants showed no inhibitory roles in centrosome over-duplication (Fig. 2c,d). To further investigate whether centriolar Cdc6 or non-centriolar/elsewhere found Cdc6 is responsible for inhibiting centrosome duplication, we overexpressed the nuclear localized Cdc6 Δcy mutant in the presence of HU and found that this mutant was incapable of inhibiting the HU-induced centrosome over-duplication (Fig. 2e,f; Supplementary Fig. 3c,d). However, a fusion protein of Cdc6 Δcy and Plk4[CTS]-domain[41], which is forcibly targeted to the centrosome, regained the ability to inhibit the HU-induced centrosome over-duplication (Fig. 2e,f; Supplementary Fig. 3c,d), indicating that the centrosomal Cdc6 inhibits centrosome over-duplication. We also tested the inhibitory role of Cdc6 on centrosome duplication in physiologically untreated cells. We overexpressed GFP-Cdc6 or GFP vector in cells for 50 h (longer than two cell cycles), while the cell cycle was not influenced by Cdc6 overexpression (Supplementary Fig. 6a,b) and counted the cells with 2, 3–4 and >4 centrioles (Fig. 2g). The results showed that there was a percentage reduction of cells with 3 or 4 centrioles among Cdc6-overexpressing cells compared to GFP-overexpressing cells (Fig. 2g), indicating that overexpressing Cdc6 restrained centrosome duplication. Taken altogether, we conclude that the centrosomal Cdc6 negatively regulates centrosome duplication.

**Cdc6 colocalizes with Plk4 and cartwheel proteins.** Thereafter, we investigated that Cdc6 is involved in which step of the regulation of centrosome duplication. Since Cdc6 is predicted to interact with Plk4 (ref. 42) (http://www.mitocheck.org), and Plk4 triggers the centriole biogenesis[17], we tested the precise centrosomal localization of Cdc6 and Plk4. Through co-expressing exogenous Cdc6 and Plk4, we found that they were co-localized at the centrosomes (Fig. 3a). To observe the co-localization more precisely, we used super-resolution three-dimensional structured illumination (3D-SIM) microscopy to detect the co-localization of Cdc6 and Plk4 in the context of a centriole distal side-localized marker CP110 (Fig. 3b). The result showed that Cdc6 colocalizes with Plk4 but not the distal side-localized centriole protein, CP110 (Fig. 3b,g). To gain higher resolution, by super-resolution stimulated emission depletion (STED) microscopy, we confirmed that endogenous Cdc6 colocalizes with endogenous Plk4 or GFP-tagged Plk4 (Fig. 3c,d). As previously reported, Plk4 localized in a dot-like manner on the procentriole formation site in S phase and colocalizes with cartwheel components Sas-6 and STIL[29]. We further investigated whether Cdc6 co-localizes with Sas-6 and STIL. Sas-6 localizes on the central hub of cartwheel and initiates centriole biogenesis[26]. STIL and Sas-6 are mutually dependent for their centriolar localization and colocalize during S and G2 phase[16,29]. We found that Cdc6 colocalized with GFP-Sas-6 under both SIM (Fig. 3b,g) and STED microscopy (Fig. 3f,g). We observed that Cdc6 also colocalized with STIL under STED microscopy (Fig. 3e,g). Taken

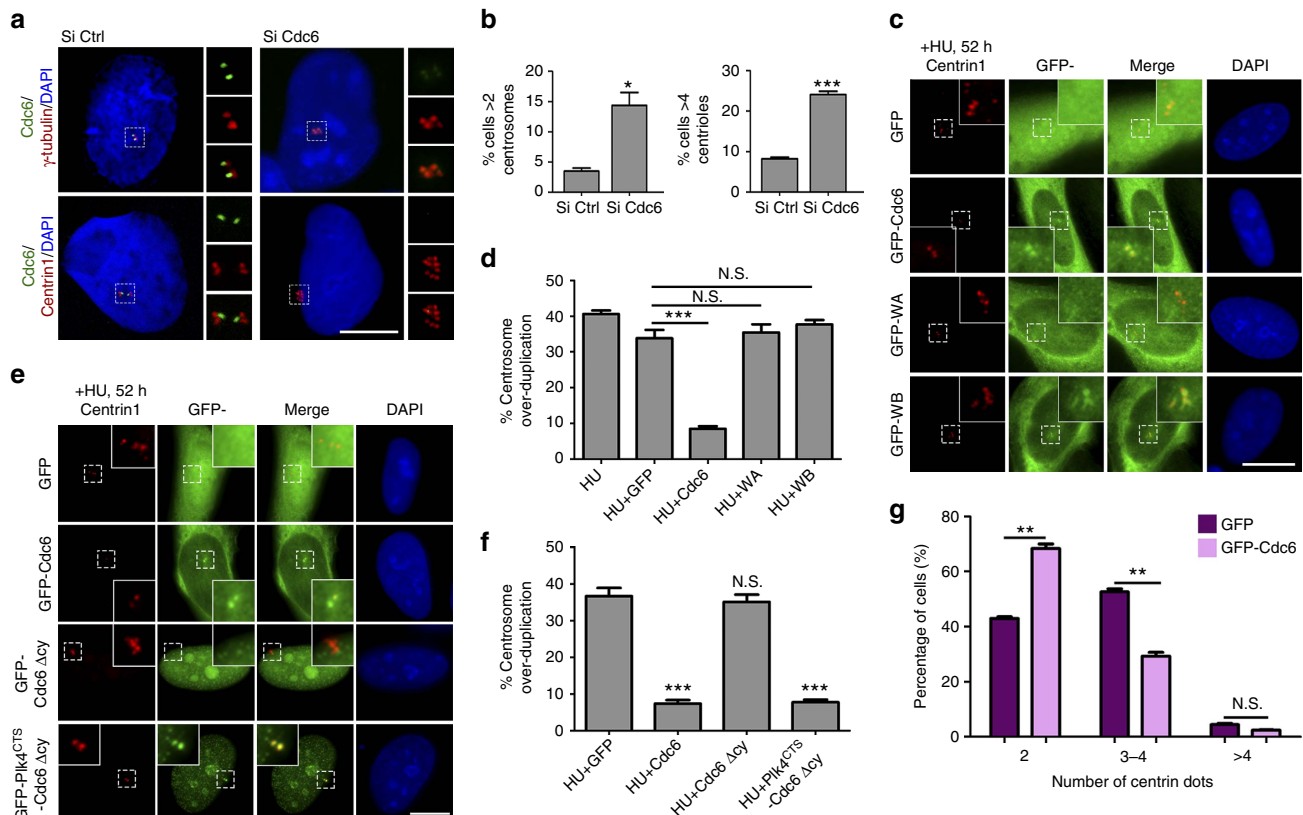

**Figure 2 | Cdc6 negatively regulates centrosome duplication. (a)** Cdc6 depletion induces centriole and centrosome over-duplication. Immunofluorescence of Cdc6 (green), centrin1 (red) and γ-tubulin (red) in U2OS cells transfected with control or Cdc6 siRNA. **(b)** Quantitation of cells with >2 centrosomes or >4 centrioles in **a**. Approximately 300 cells were counted per sample, and three independent experiments were conducted. **(c)** Cdc6 WT, but not WA or WB mutant, inhibits HU-induced centrosome over-duplication. U2OS cells were transfected with GFP, GFP-Cdc6, GFP-WA or GFP-WB, and treated with 16 mM HU for 52 h to allow centriole amplification. The cells were then stained with a centrin1 antibody. **(d)** Quantitation of cells with >4 centrioles in **c**. Approximately 300 cells were counted per sample, and three independent experiments were conducted. **(e)** Centrosomal localization of Cdc6 is required for its inhibition on the HU-induced centrosome over-duplication. U2OS cells were transfected with GFP, GFP-tagged Cdc6, GFP-tagged Cdc6 Δcy or GFP-tagged Plk4$^{CTS}$-Cdc6 Δcy, and treated with 16 mM HU for 52 h to allow centriole amplification. The cells were then stained with a centrin 1 antibody. **(f)** Quantitation of cells with >4 centrioles in **e**. Approximately 100 cells were counted per sample, and three independent experiments were conducted. **(g)** Overexpression of Cdc6 inhibits centriole duplication during the normal cell cycle. U2OS cells were transiently transfected with GFP or GFP-tagged Cdc6 for 50 h and stained with a centrin1 antibody. The number of centrin1-positive dots in transfected cells was counted. Approximately 300 cells were counted per sample, and three independent experiments were conducted. The statistical data in (**b,d,f,g**) are presented as means ± s.d. *$P<0.05$, **$P<0.01$, and ***$P<0.001$; N.S., no significant difference (Student's *t*-test). DNA was stained with DAPI. Scale bars, 10 μm. Insets in **a,c** and **e** are high-magnification views of the regions indicated in the low-magnification images.

together, these results demonstrate that Cdc6 colocalizes with Plk4, Sas-6 and STIL at the centriole during S phase, indicating a functional role of Cdc6 in procentriole biogenesis.

**Plk4 binds Cdc6 in S phase and phosphorylates Cdc6.** To confirm the predicted interaction between Cdc6 and Plk4 (ref. 42) (http://www.mitocheck.org), we detected and confirmed the direct interaction between Plk4 and Cdc6 *in vitro* by protein–protein interaction assay using purified GST-Plk4 and His-Cdc6 protein (Fig. 4b) and pull-down assay (Supplementary Fig. 4b). We also showed the interaction between endogenous Cdc6 and Plk4 *in vivo* by co-IP with Cdc6 antibody (Fig. 4a). The interaction between overexpressed Cdc6 and Plk4 was also confirmed by reciprocal co-IP using GFP-Cdc6 or GFP-Plk4 as the bait (Fig. 4c; Supplementary Fig. 4a). However, the kinase-dead mutant Plk4 K41M does not bind to Cdc6, although the Cdc6 protein level was not affected by Plk4 K41M mutant over-expression (Fig. 4c; Supplementary Fig. 4c), indicating that the kinase activity of Plk4 is required for the interaction between

Cdc6 and Plk4. We further examined whether the interaction between Cdc6 and Plk4 is cell cycle dependent. By co-IP in HEK293 cells co-transfected with GFP-Plk4 and Myc-Cdc6 and synchronized to the G1/S transition, S or G2 phase, we found that the interaction between Plk4 and Cdc6 was limited to S phase (Fig. 4d), suggesting that a functional interplay between Plk4 and Cdc6 exists only during S phase. To determine which region on Cdc6 binds to Plk4, through co-IP in HEK293 cells co-transfected with GFP-tagged Cdc6 truncates and Myc-tagged Plk4, we observed that the N-terminal truncates amino acids 1–179 and 179–407 of Cdc6, but not the C-terminal truncate amino acids 407–560, interacted with Plk4 (Fig. 4e). Taken together, these data demonstrate that Cdc6 binds the active Plk4 kinase via N-terminus of Cdc6 during S phase.

As the kinase-active WT Plk4, but not the kinase-dead mutant, interacts with Cdc6, we further examined whether Cdc6 is phosphorylated by Plk4. By using GPS 3.0 (Group-based Prediction System, version 3.0) for Plk4 phosphorylation sites prediction, we found that serine 30 and threonine 527 on Cdc6 showed highest scores as Plk4 phosphorylation sites

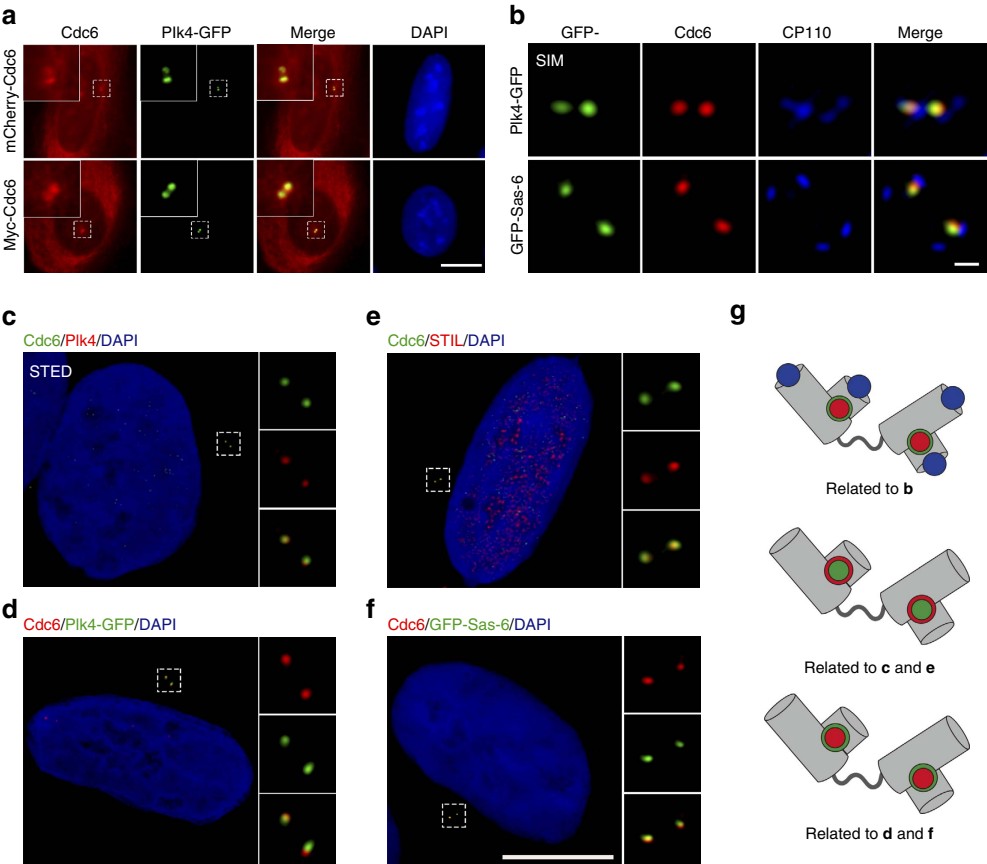

**Figure 3 | Cdc6 colocalizes with Plk4 and cartwheel proteins Sas-6 and STIL during S phase.** (**a**) Cdc6 colocalizes with Plk4. U2OS cells were co-transfected with mCherry-tagged Cdc6 (upper panel) or Myc-tagged Cdc6 (lower panel) with GFP-tagged Plk4. Myc-tagged Cdc6 was stained with a Myc antibody. (**b**) Cdc6 colocalizes with Plk4 and Sas-6 viewed by 3D-SIM. 3D-SIM images of U2OS cells transfected with GFP-tagged Plk4 (upper panel) or GFP-tagged Sas-6 (lower panel) and stained for endogenous Cdc6 and CP110. (**c–f**) Cdc6 colocalizes with Plk4, STIL and Sas-6 viewed by STED. Immunofluorescence of endogenous Cdc6 (green) and Plk4 (red), or Cdc6 (green) and STIL (red) in U2OS cells (**c,e**). Immunofluorescence of endogenous Cdc6 (red) in U2OS cells transfected with GFP-tagged Plk4 or GFP-tagged Sas-6 (**d,f**). (**g**) Schematic diagrams illustrating the localization of proteins at the centrosome in **b–f**. Different proteins are indicated by the colours of staining in **b–f**. DNA was stained with DAPI. Scale bars in **a,c–f**, 10 μm. Scale bar in **b**, 0.2 μm. Insets in **a,c–f** are high-magnification views of the regions indicated in the low-magnification images.

(Supplementary Fig. 4d)[43]. We first tested in cells the phosphorylation of Cdc6 WT, a single mutation of serine 30 to alanine (S30A), a single mutation of threonine 527 to alanine (T527A) and a double mutation of S30A and T527A (denoted as 2A) mutants using Phos-Tag acrylamide assay. As shown in Fig. 4f, the slower mobility shift band represents phosphorylated Cdc6, the faster mobility shift band represents unphosphorylated Cdc6. Cdc6 S30A or T527A mutant partially decreased Cdc6 phosphorylation compared to Cdc6 WT, and the Cdc6 2A double alanine mutant further decreased Cdc6 phosphorylation (Fig. 4f). This result suggests that Cdc6 is phosphorylated on S30 and T527 in cells. Next, we performed an *in vitro* kinase assay by incubating purified Plk4 and Cdc6 proteins in the presence of $[\gamma\text{-}^{32}\text{P}]$-ATP. The result revealed that Cdc6 was phosphorylated by Plk4, and S30A mutant or T527A mutant of Cdc6 reduced, but not completely abolished, Cdc6 phosphorylation, while the double mutation 2A mutant abolished the phosphorylation of Cdc6 by Plk4 (Fig. 4g,h; Supplementary Fig. 4e–g). Taken together, these results demonstrate that Plk4 binds and phosphorylates Cdc6 at serine 30 and threonine 527.

**Cdc6 2D mutation cannot inhibit centrosome duplication.** To explore the role of Cdc6 phosphorylation by Plk4 on centrosome duplication, we compared the ability of Cdc6 WT, 2A mutant or a double mutation of S30D and T527D (denoted as 2D) mutant in suppressing the HU-induced centrosome over-duplication (Fig. 5a,b; Supplementary Fig. 5a,b). We found that the unphosphorylatable Cdc6 2A mutant efficiently inhibited the HU-induced centrosome over-duplication similar to WT Cdc6. By contrast, the phosphorylation-mimic Cdc6 2D mutant was deficient in inhibiting the HU-induced centrosome over-duplication similar to the negative control of GFP vector (Fig. 5a,b; Supplementary Fig. 5a,b). The HU-induced S phase arrest in cell cycle was not influenced by Cdc6 WT, 2A or 2D mutant over-expression (Supplementary Fig. 5f,g). These results suggested that unphosphorylated Cdc6 inhibits centrosome over-duplication, whereas its phosphorylation by Plk4 abolishes the inhibitory role. We further analysed centrosome duplication in physiologically untreated cells expressing GFP-Cdc6 WT, 2A or 2D mutant for 50 h under conditions in which the cell cycle was not influenced by overexpression (Supplementary Fig. 6a,b). By quantifying the proportions of cells with 2, 3–4 or over 4 centrioles after Cdc6 expression for 50 h (longer than two cell cycles), we also found that 2A efficiently inhibited centrosome duplication similar to WT Cdc6 but not 2D mutant (Fig. 5c). Taken together, we conclude that phosphorylation of Cdc6 by Plk4 abolishes its inhibitory role on centrosome duplication.

As Plk4 overexpression leads to centriole amplification[13], and Cdc6 suppresses centriole amplification as discussed above, we

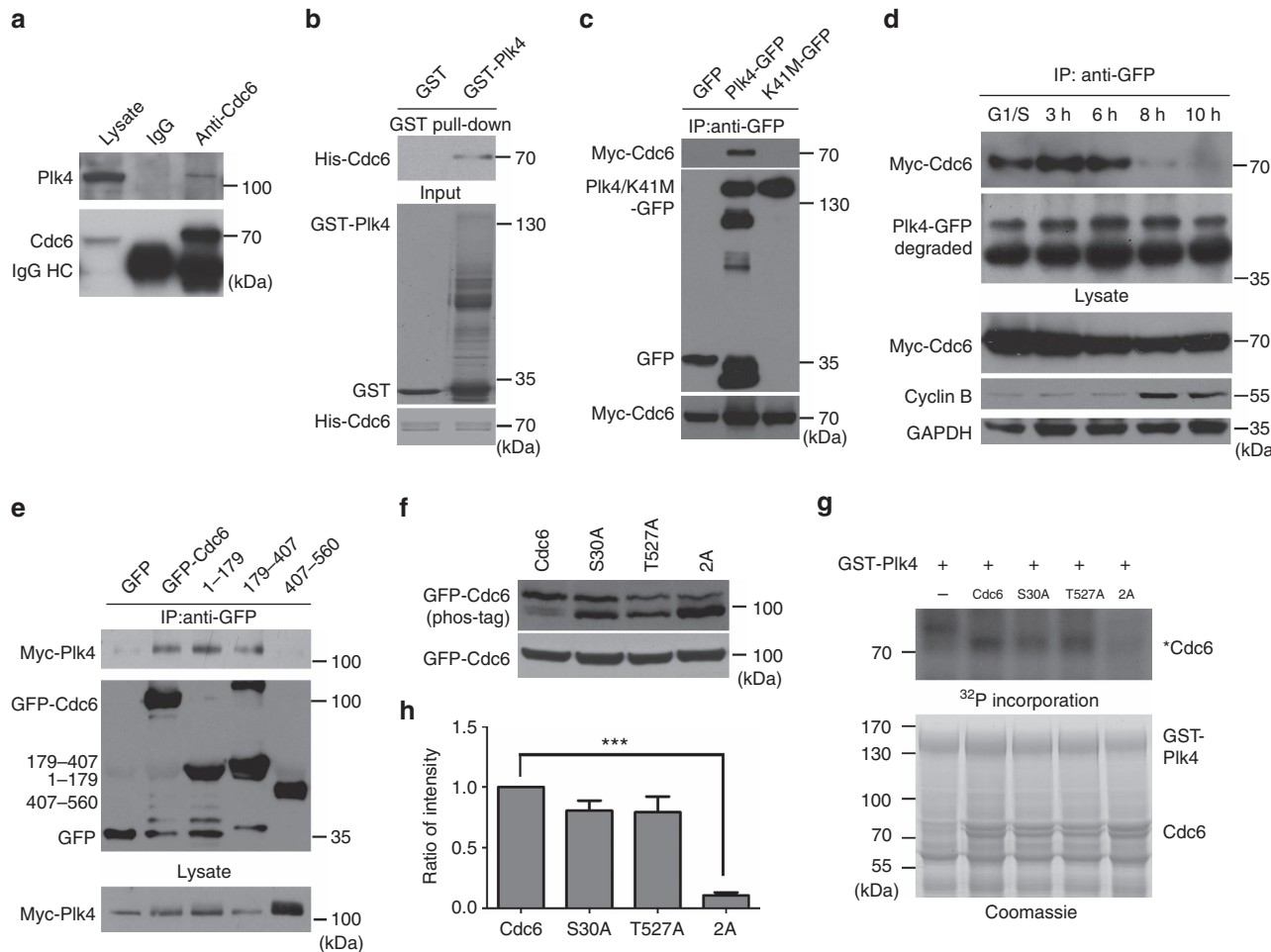

**Figure 4 | Plk4 binds Cdc6 in S phase and phosphorylates Cdc6.** (**a**) Endogenous Cdc6 interacts with Plk4 in cells. HEK293 total cell extract was immunoprecipitated with a Cdc6 antibody, and probed with Cdc6 and Plk4 antibodies. (**b**) Plk4 directly binds Cdc6 *in vitro*. Sepharose beads coupled with purified GST or GST-Plk4 were incubated with purified His-Cdc6 and analysed using His antibody. The loadings of indicated proteins are shown by Coomassie blue staining. (**c**) The Plk4 kinase-dead mutant K41M does not bind Cdc6 in cells. HEK293 cells were co-transfected with Myc-Cdc6 and GFP-Plk4 WT or K41M mutant. Total cell extracts were immunoprecipitated with GFP antibody, and probed with Myc and GFP antibodies. (**d**) Cdc6 interacts with Plk4 during S phase. HEK293 cells were co-transfected with GFP-Plk4 and Myc-Cdc6 and synchronized at the G1/S transition. The cells were then released for the indicated time period, collected for immunoprecipitation using GFP antibody, and analysed by western blotting using Myc, GFP, cyclin B and GAPDH antibodies. (**e**) N-terminus of Cdc6, but not C-terminus of Cdc6, interacts with Plk4. GFP, GFP-tagged Cdc6 WT or Cdc6 truncate was immunoprecipitated with GFP antibody from total cell extract of HEK293 cells co-transfected with Myc-Plk4. Immunoprecipitated proteins were analysed using Myc and GFP antibodies. (**f**) Cdc6 is phosphorylated on S30 and T527 in cells. HEK293 cells were transfected with GFP-tagged Cdc6 or Cdc6 mutants, and the cell lysates were probed with GFP antibody using Phos-Tag acrylamide assay (upper panel). (**g**) Plk4 phosphorylates Cdc6 on S30 and T527 *in vitro*. Purified GST-Plk4 was incubated with purified His-tagged Cdc6 or Cdc6 mutant proteins in the presence of [γ-$^{32}$P]-ATP, followed by autoradiography. Coomassie blue staining shows the loading of indicated proteins. Asterisk indicates the phosphorylated Cdc6. (**h**) Quantitation of (**g**) for Cdc6 WT, S30A, T527A or 2A phosphorylation from four independent experiments. The relative intensity of the phosphorylation signal of each protein was normalized to its protein level by ImageJ. The phosphorylation signal intensity of Cdc6 WT was arbitrarily set at an intensity of 1.0. The statistical data in **h** is presented as means ± s.d. ***$P < 0.001$ (Student's *t*-test).

tested whether Plk4-mediated centriole amplification could be suppressed by Cdc6 (ref. 17). To this end, mCherry-tagged Cdc6 WT, 2A or 2D mutant was co-expressed with GFP-Plk4 in U2OS cells, and the proportions of cells with centriole amplification were analysed. Compared to 52.7% of the control cells co-expressing mCherry and Plk4 showing the centriole amplification, only 26.6% of the cells co-expressing mCherry-Cdc6 WT and Plk4 exhibited the centriole amplification (Fig. 5d,e), indicating that Cdc6 overexpression significantly suppressed the Plk4-induced centriole amplification (Student's *t*-test). Moreover, we found that 2A efficiently suppressed the Plk4-induced centriole amplification similar to WT Cdc6, but not 2D mutant (Fig. 5d,e), consistent with the scenario that

unphosphorylated Cdc6, rather than the phosphorylated, inhibits centrosome duplication (Fig. 5a–c). We also tested and confirmed that the cell cycle was not influenced by co-expression of mCherry-tagged Cdc6 WT, 2A or 2D mutant with GFP-Plk4 in U2OS cells (Supplementary Fig. 6c,d). To test whether Cdc6 inhibits centrosome duplication by inhibiting Plk4, we performed double knockdown of Cdc6 and Plk4 (Supplementary Fig. 5e). The result showed that simultaneous depletion of Cdc6 and Plk4 restored the centrioles numbers in mitosis which was decreased after Plk4 depletion only (Supplementary Fig. 5c,d), indicating that Cdc6 is not a Plk4 inhibitor, and that Cdc6 inhibits the centrosome duplication downstream or parallel of Plk4. Taken together, we conclude that phosphorylation of Cdc6 by Plk4

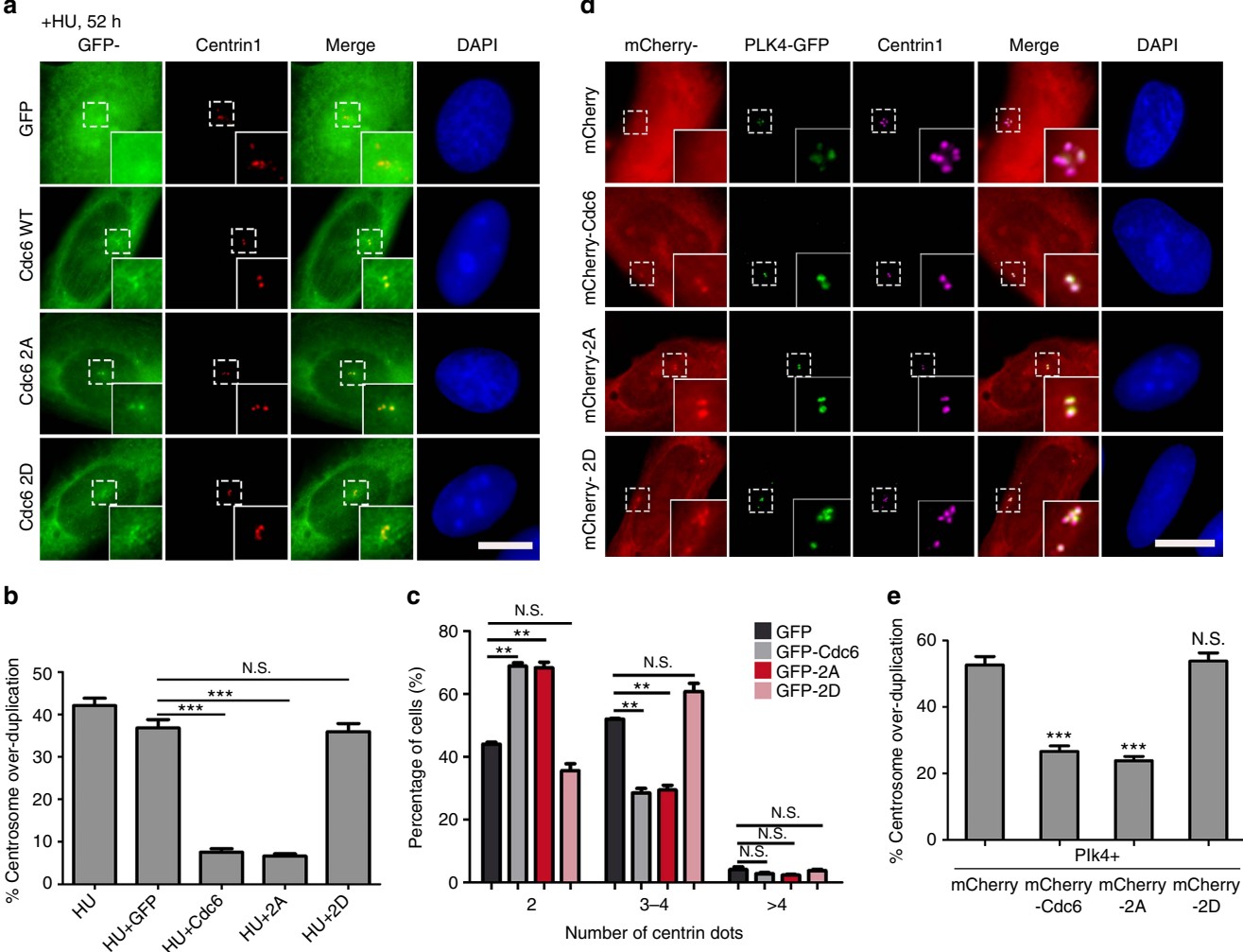

**Figure 5 | Phosphorylation of Cdc6 by Plk4 abolishes the inhibitory role of Cdc6 on centrosome duplication. (a)** Cdc6 WT or 2A mutant, but not 2D mutant, inhibits HU-induced centrosome over-duplication. U2OS cells were transfected with GFP, GFP-tagged Cdc6 WT, GFP-tagged Cdc6 2A or 2D mutant, and treated with 16 mM HU for 52 h to allow centriole amplification. The cells were then stained with a centrin1 antibody. (**b**) Quantitation of cells with >4 centrioles in **a**. Approximately 100 cells were counted per sample, and three independent experiments were conducted. (**c**) Overexpression of Cdc6 WT or 2A mutant, but not 2D mutant, inhibits the centrosome duplication during the cell cycle. U2OS cells were transfected with GFP, GFP-tagged Cdc6 WT and GFP-tagged Cdc6 2A or 2D mutant for 50 h. The cells were then stained with a centrin1 antibody. The number of centrin1-positive dots in transfected cells was counted. Approximately 300 cells were counted per sample, and three independent experiments were conducted. (**d**) Overexpression of Cdc6 WT or 2A mutant, but not 2D mutant, suppresses Plk4-induced centriole amplification. U2OS cells were transfected with GFP-tagged Plk4 and mCherry-tagged Cdc6 WT, 2A or 2D mutant for 40 h. The cells were then stained with a centrin1 antibody. (**e**) Quantitation of cells with >4 centrioles in **d**. Approximately 100 cells were counted per sample, and three independent experiments were conducted. The statistical data in **b,c** and **e** are presented as means ± s.d. **$P < 0.01$, and ***$P < 0.001$; N.S., no significant difference (Student's $t$-test). DNA was stained with DAPI. Scale bars, 10 µm. Insets in **a** and **d** are high-magnification views of the regions indicated in the low-magnification images.

attenuates its function in inhibiting centrosome duplication and centrosome over-duplication induced by HU treatment or Plk4 overexpression.

**Cdc6 restrains centrosome duplication via inhibiting Sas-6.** Plk4 triggers the centriole biogenesis through phosphorylating its substrates, such as the cartwheel protein STIL[28,29] and the F-box protein Fbxw5 (refs 28,29,31), and facilitating the loading of Sas-6 to the cartwheel. As Plk4 phosphorylates Cdc6 and Cdc6 colocalizes with Sas-6 on the cartwheel (Fig. 3b,f), we tested the relationship between Cdc6 phosphorylation and Sas-6. First, we investigated whether Cdc6 interacts with Sas-6. By reciprocal co-IP using Cdc6 or Sas-6 as bait, we found that endogenous Cdc6 specifically interacted with endogenous Sas-6 but not other

centrosome proteins CPAP, Cep164 or STIL (Fig. 6a), excluding the possibility that Cdc6 interacts with Sas-6, Plk4 or cyclin A simply by pulling down the entire centrosome. The interaction between overexpressed Cdc6 and Sas-6 was also confirmed by reciprocal co-IP using GFP-Cdc6 or GFP-Sas-6 as the bait (Supplementary Fig. 7a,b). We also performed *in vitro* protein–protein binding assay using *in vitro* transcribed and translated Cdc6 and Sas-6 proteins (Fig. 6b). His-Cdc6 was then immunoprecipitated by Cdc6 antibody, and the Cdc6-bound His-Sas-6 was detected. This result showed that Cdc6 directly binds with Sas-6 *in vitro* (Fig. 6b).

Sas-6 forms the backbone of the cartwheel and its depletion inhibits many forms of centriole amplification[12]. We first tested whether Sas-6 depletion inhibits Cdc6 depletion induced centriole amplification. We performed double knockdown of Cdc6 and

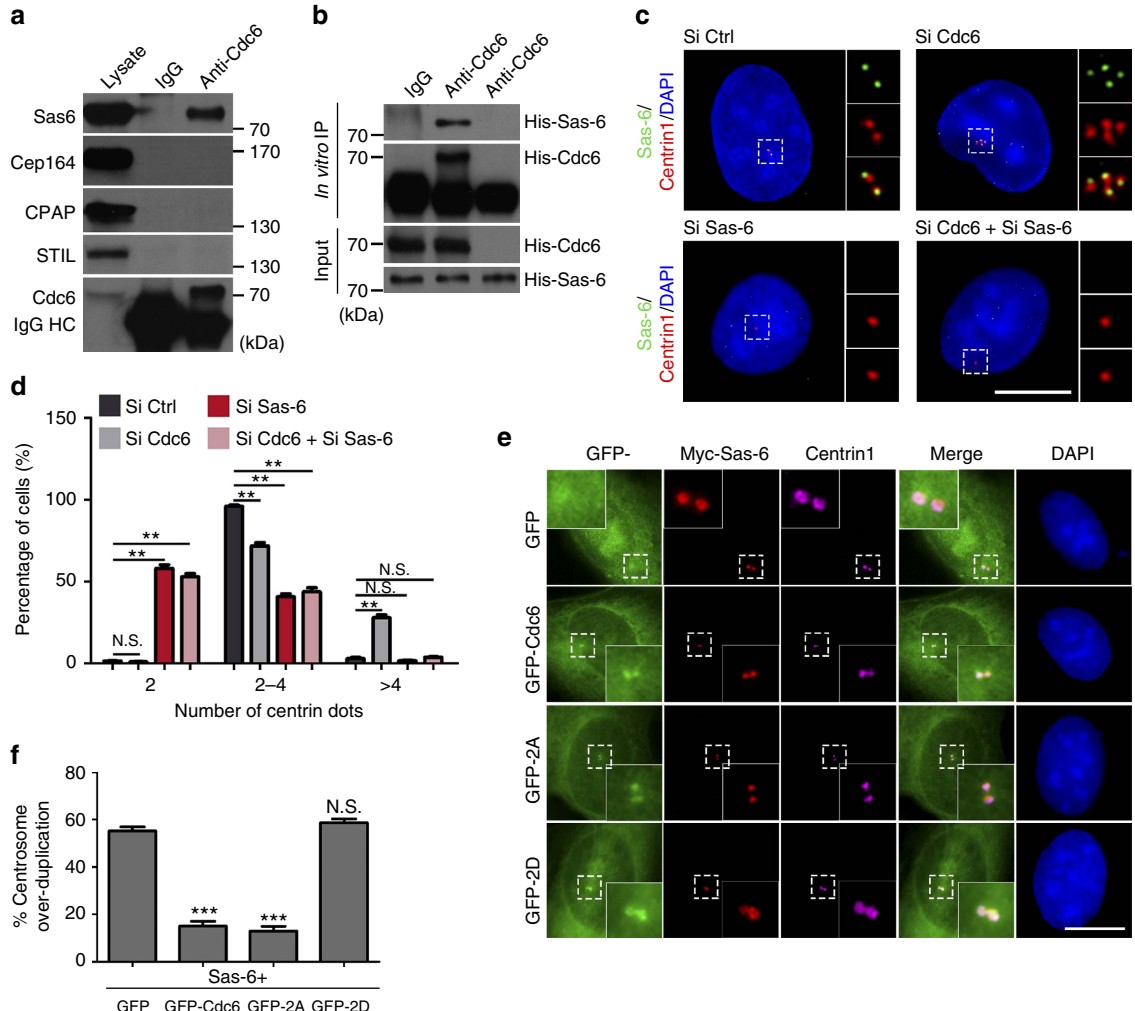

**Figure 6 | Cdc6 restrains centrosome duplication via inhibiting Sas-6 in a Plk4 phosphorylation regulated way.** (**a**) Endogenous Cdc6 interacts with Sas-6 in cells. HEK293 total cell extract was immunoprecipitated with a Cdc6 antibody, and probed with Cdc6, Sas-6, Cep164, CPAP and STIL antibodies. (**b**) Cdc6 directly binds Sas-6 in vitro. In vitro transcribed and translated His-tagged Sas-6 was incubated with or without in vitro transcribed and translated His-tagged Cdc6. His-Cdc6 was then immunoprecipitated with a Cdc6 antibody. The bound proteins were analysed with Cdc6 and Sas-6 antibodies. (**c**) Simultaneous depletion of Sas-6 eliminates the Cdc6 depletion-induced centriole amplification. U2OS cells were transfected with control siRNA, Cdc6 siRNA or Sas-6 siRNA; or co-transfected with Cdc6 siRNA and Sas-6 siRNA. The cells were then stained with centrin1 (red) and Sas-6 (green) antibodies to identify the cells with efficient depletion of Sas-6. (**d**) Quantitation of cells with indicated centrin1-positive dots number in **c**. Approximately 100 cells were counted per sample, and three independent experiments were conducted. (**e**) Cdc6 WT or Cdc6 2A mutant, but not Cdc6 2D mutant, suppresses the Sas-6-induced centriole amplification. Myc-tagged Sas-6 was co-transfected with GFP, GFP-tagged Cdc6 WT or GFP-tagged Cdc6 2A or 2D mutant for 28 h. The cells were then stained with Myc and centrin1 antibodies. (**f**) Quantitation of cells with > 4 centrioles in **e**. Approximately 100 cells were counted per sample, and three independent experiments were conducted. The statistical data in **d** and **f** are presented as means ± s.d. **$P < 0.01$, and ***$P < 0.001$; N.S., no significant difference (Student's $t$-test). DNA was stained with DAPI. Scale bars, 10 µm. Insets in **c** and **e** are high-magnification views of the regions indicated in the low-magnification images.

Sas-6 and found that the simultaneous knockdown of Sas-6 and Cdc6 eliminated the Cdc6 depletion-induced centriole amplification, and even resulted in large percentage of cells with single centriole similar to Sas-6 knockdown (Fig. 6c,d; Supplementary Fig. 7e). These results suggested that Sas-6, as an essential centrosome duplication initiator, its depletion overrides the Cdc6 depletion-induced centriole amplification. Since it has been reported that Sas-6 overexpression induces centriole amplification[14,44], we co-overexpressed GFP-Cdc6 and Myc-Sas-6 in cells to test whether Cdc6 negatively regulates centriole amplification through inhibiting Sas-6. The results showed that, co-overexpressing Cdc6 WT with Sas-6 inhibited centriole amplification induced by overexpressing Sas-6 (Fig. 6e,f; Supplementary Fig. 7c,d), and that Cdc6 2A mutant, but not

Cdc6 2D mutant, suppressed the Sas-6-induced centriole amplification similar to WT Cdc6 (Fig. 6e,f; Supplementary Fig. 7c,d).

**Cdc6 phosphorylation by Plk4 disrupts Cdc6-Sas-6 interaction.** To investigate how Plk4-mediated phosphorylation suppresses the inhibition of Cdc6 on Sas-6, we first depleted Plk4 by Plk4 siRNA and found this impaired the interaction between Cdc6 and Sas-6, revealing that Plk4 is required for the interaction between Cdc6 and Sas-6 (Fig. 7a; Supplementary Fig. 7f). Next, we tested whether the phosphorylation of Cdc6 by Plk4 regulates the interaction between Sas-6 and Cdc6. By co-IP, we found that the interaction between Sas-6 and Cdc6 2A mutant was stronger than

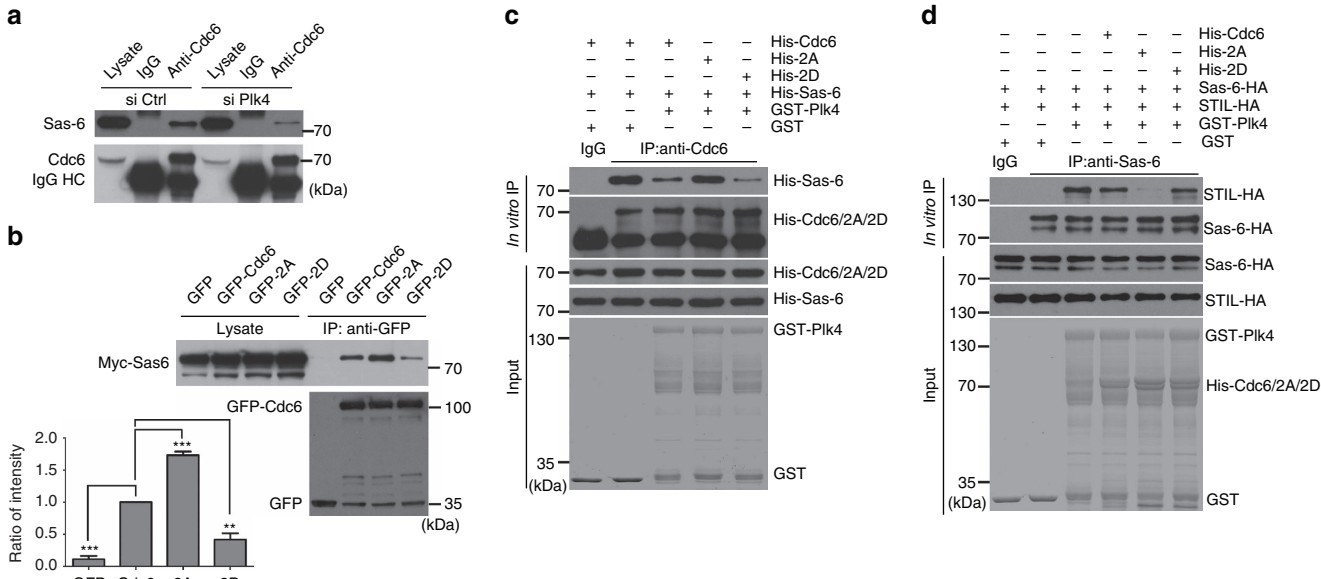

**Figure 7 | Cdc6 phosphorylation by Plk4 disrupts the Cdc6-Sas-6 interaction and facilitates Sas6-STIL complex formation. (a)** The interaction between Cdc6 and Sas-6 is impaired by Plk4 depletion in cells. The whole cell extract of HEK 293 T cells transfected with control siRNA or Plk4 siRNA was immunoprecipitated with a Cdc6 antibody, and probed with Cdc6 and Sas-6 antibodies. (**b**) Cdc6 phosphorylation inhibits the interaction between Cdc6 and Sas-6 in cells. HEK293 cells co-transfected with Myc-tagged Sas-6 and GFP-tagged Cdc6, 2A or 2D mutant were immunoprecipitated with a GFP antibody, and analysed by western blotting using Myc and GFP antibodies. Quantitation of the interaction between Sas-6 and Cdc6 WT, 2A or 2D mutant was analysed from three independent experiments. The relative interaction between Sas-6 and Cdc6 was normalized to the immunoprecipitated Cdc6 by ImageJ. The interaction between Myc-Sas-6 and GFP-Cdc6 WT was arbitrarily set at an intensity of 1.0. (**c**) Cdc6 phosphorylation inhibits the interaction between Cdc6 and Sas-6 in vitro. His-tagged Cdc6, 2A or 2D mutant protein transcribed and translated in vitro was incubated with or without Plk4 kinase in the presence of in vitro transcribed and translated His-Sas-6 in kinase reaction buffer. His-Cdc6 was then immunoprecipitated by Cdc6 antibody, and the bound proteins were analysed with Cdc6 and Sas-6 antibodies. The loadings of GST and GST-tagged Plk4 proteins are shown by Coomassie blue staining. (**d**) Cdc6 phosphorylation by Plk4 suppresses the inhibition of Cdc6 on Sas6-STIL complex formation. HA-tagged Sas-6 protein transcribed and translated in vitro was incubated with indicated purified proteins in the presence of in vitro transcribed and translated HA-tagged STIL in kinase reaction buffer. HA-Sas-6 was then immunoprecipitated by Sas-6 antibody, and the bound proteins were analysed with Sas-6 and STIL antibodies. The loadings of GST, GST-tagged Plk4, His tagged Cdc6, 2A and 2D mutant proteins are shown by Coomassie blue staining. The statistical data in **b** is presented as means ± s.d. **P < 0.01 and ***P < 0.001 (Student's t-test).

that between Sas-6 and Cdc6 WT or Cdc6 2D mutant, suggesting that Cdc6 phosphorylation by Plk4 negatively regulates the interaction between Cdc6 and Sas-6 (Fig. 7b; Supplementary Fig. 7g,h). To confirm this in vitro, His-tagged Cdc6, 2A mutant or 2D mutant protein transcribed and translated in vitro was incubated with or without Plk4 kinase in the presence of His-Sas-6 in kinase reaction buffer (Fig. 7c). His-Cdc6 was then immunoprecipitated by Cdc6 antibody and the Cdc6-bound His-Sas-6 was analysed (Fig. 7c). Consistent with the in vivo co-IP experiment, both phosphorylated Cdc6 (incubated with Plk4) and phosphorylation-mimic 2D mutant inhibited the interaction between Cdc6 and Sas-6 (Fig. 7c), whereas unphosphorylated Cdc6 (without incubation with Plk4) and unphosphorylatable Cdc6 2A mutant had a strong interaction with Sas-6 (Fig. 7c).

We further investigated how Cdc6 phosphorylation regulates centrosome duplication through Sas-6. It was reported that Sas-6 interacts with another procentriole formation core protein STIL in a Plk4 phosphorylation-dependent manner on procentriole formation site to induce centrosome duplication[15,29]. Herein we tested whether Cdc6 phosphorylation regulates the stability of the interaction between Sas-6 and STIL (Fig. 7d). Through an in vitro protein–protein binding assay, we showed that only phosphorylated STIL could interact with Sas-6 (Fig. 7d; Supplementary Fig. 7i), and the interaction between Sas-6 and STIL is regulated by Cdc6 phosphorylation. Both the phosphorylated Cdc6 (incubated with Plk4) and the phosphorylation-mimic Cdc6 2D mutant facilitated a strong

interaction between Sas-6 and STIL, while the unphosphorylatable Cdc6 2A mutant inhibited the interaction between Sas-6 and STIL (Fig. 7d), indicating that Cdc6 phosphorylation by Plk4 promotes the release of Sas-6 from its inhibition and results in the stable interaction between Sas-6 and STIL.

Taken altogether, we demonstrated that Cdc6 negatively regulates the centriole duplication by inhibiting the interaction between Sas-6 and STIL, and that this inhibitory role of Cdc6 on centriole duplication is negatively regulated by Plk4 phosphorylation. Cdc6 phosphorylation by Plk4 disrupts the binding of Sas-6 to Cdc6, facilitates the interaction between Sas-6 and STIL, and thus centrosome duplication.

## Discussion

Centrosome duplication is strictly regulated, occurring only once per cell cycle to avoid multipolar spindle assembly and genome instability[45]. Altogether with centrosome duplication during the cell cycle, DNA replication also occurs only once per cell cycle via the pre-RC licensing mechanism[6,7]. Three DNA replication pre-replicative complex (pre-RC) components, Orc1, MCM5 and geminin, have been reported to restrain centrosome over-duplication[8–11,46], indicating that a linkage exists between the initiation of centrosome duplication and DNA replication. However, the functional association of DNA replication initiators with centrosomal proteins involved in centrosome duplication is still unknown. In this study, we identified DNA

replication initiator Cdc6 as a substrate of Plk4 kinase. Cdc6 negatively regulates centrosome duplication by binding and inhibiting Sas-6 from forming a stable complex with STIL, and this inhibitory function on centrosome duplication was antagonistically regulated by Plk4 phosphorylation during S phase. The identification of Cdc6 as a substrate of Plk4 kinase and the inhibition of Cdc6 on the Sas-6-STIL interaction, as well as the regulation of Cdc6-Sas-6 interactions by Cdc6 phosphorylation, prompted us to further explore the underlying mechanism of centrosome duplication and the involvement of DNA replication initiators in this process in future studies.

On the basis of our present and previous reported results, we propose a working model to depict the function of Cdc6 in centrosome duplication (Fig. 8). During G1 phase, Plk4 localizes in a ring-like manner surrounding the parental centrioles, while the procentriole formation core protein Sas-6 and STIL are degraded[16,44]. Cdc6 localizes in the nucleus and is absent at the centrosomes during G1 phase. Centrosome duplication cannot initiate at this time (Fig. 8a). When cells progress to S phase, Cdc6 is recruited to the centrosomes by cyclin A to inhibit untimely centrosome duplication. To trigger centrosome duplication, Plk4 phosphorylates Cdc6 and thus attenuates the inhibitory role of Cdc6 on Sas-6, facilitating the interaction between Sas-6 and STIL, and triggering centrosome duplication during S phase (Fig. 8b). Constant phosphorylation of Cdc6 by Plk4 abolishes its inhibitory activity on Sas-6 to allow Sas-6 to initiate centrosome over-duplication (Fig. 8d). By contrast, unphosphorylated Cdc6 exerts a stronger inhibitory role on Sas-6 and attenuates the interaction between Sas-6 and STIL, and thus restrains the procentriole growth (Fig. 8c). Therefore, to ensure normal centrosome duplication during the cell cycle, the expression and activity of both Plk4 and Cdc6 must be tightly balanced to prevent over-duplication of the centrosome. In summary, our results reveal a novel mechanism in which the opposing activities of Plk4 and Cdc6 ensure the centrosome duplicate once per cell cycle via regulating the interaction of Sas-6 with STIL.

Several other proteins have been reported to be involved in controlling centrosome amplification. Cdc14B associates with the centrosomes and inhibits centriole amplification in S phase-arrested cells[47]. Cep76, a centriolar protein, limits centrosome duplication to once per cell cycle[48]. PIPKIγ restrains centriole duplication by interacting with Plk4 and negatively regulating its kinase activity[49]. Cdkn3 prevents Mps1 kinase-mediated centrosome over-duplication by regulating Mps1 protein stability[50]. In addition, KLF14 represses centrosome amplification by transcriptionally inhibiting the Plk4 protein level[51]. The DNA replication pre-replicative complex (pre-RC) components, Orc1, MCM5 and geminin, restrain centrosome over-duplication in S phase-arrested cells[8–11,46]. Orc1 localizes to the centrosome via cyclin A and prevents cyclin E-dependent re-duplication of the centrosome[8]. However, the downstream targets of MCM5 and geminin are unknown. In this study, we demonstrate that the DNA replication pre-RC component Cdc6 is a Plk4 kinase substrate that restrains centrosome duplication by binding and inhibiting Sas-6 from interacting with STIL. The significance of our findings is three-fold. First, we identified a new regulatory member that restrains centrosome duplication. Second, we revealed a DNA replication pre-RC component Cdc6, in addition to Orc1, MCM5 and geminin, that represses centrosome duplication, thus reinforcing the association of DNA replication with centrosome duplication. Third, and the most important, our results enhances the mechanistic understanding of the roles of DNA replication regulators in centrosome duplication. These findings that Cdc6 takes a negative role in centrosome duplication and that Plk4 phosphorylates Cdc6 and suppresses the inhibitory role of Cdc6 establish an interplay between DNA replication initiators and centrosome duplication trigger proteins. Moreover, we have identified Cdc6 as a novel negative regulator on Sas6-STIL interaction that is regulated by Plk4, indicating Plk4 had a dual role on promoting the onset of procentriole formation. Our results also provide insights for further studies on the underlying mechanisms that prevent centrosome over-duplication and DNA

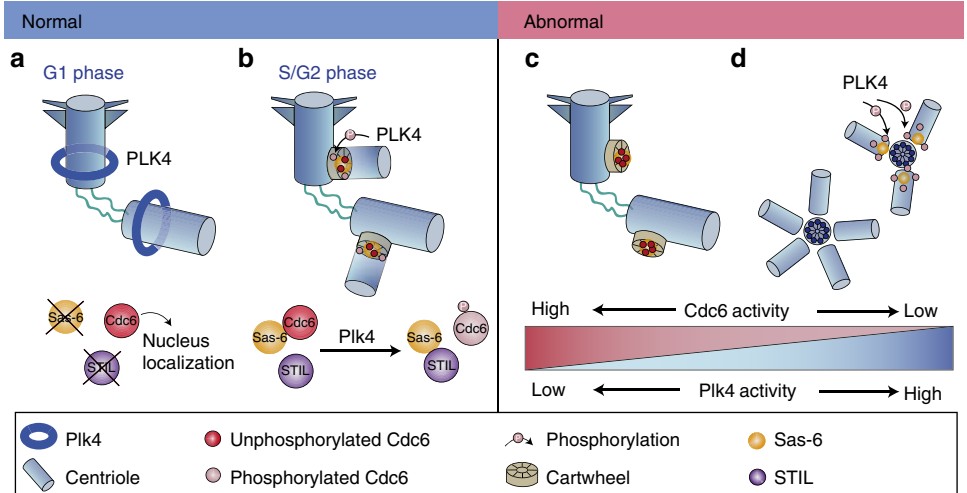

**Figure 8 | Model elucidating that Cdc6 and Plk4 antagonistically regulate centrosome duplication.** During G1 phase, Plk4 localizes in a ring-like manner surrounding the parental centrioles, while the procentriole formation core protein Sas-6 and STIL are degraded. Cdc6 localizes in the nucleus and is absent at the centrosomes during G1 phase. Centrosome duplication cannot initiate at this time (**a**). When the cells progress into S phase, Cdc6 is recruited to the cartwheel of the centrosome by cyclin A to inhibit untimely centrosome duplication. To trigger centrosome duplication during S phase, Plk4 gradually phosphorylates Cdc6 and attenuates the interaction of Cdc6 with Sas-6, facilitating Sas-6 to form a stable complex with STIL, which then triggers centrosome duplication (**b**). Unphosphorylated Cdc6 exerts a stronger inhibitory role on Sas-6 and attenuates the interaction between Sas-6 and STIL, and thus restrains the procentriole growth (**c**). By contrast, constant phosphorylation of Cdc6 by Plk4 abolishes its inhibitory activity on Sas-6 to allow Sas-6 forming a stable complex with STIL to initiate the centrosome over-duplication (**d**). Therefore, the balance of Cdc6 and Plk4 activities is crucial for the normal centrosome duplication. In summary, we propose that Cdc6 is a novel negative regulator of centrosome duplication, controlling the timely and proper centrosome duplication during the cell cycle.

re-duplication during the cell cycle. In conclusion, this work has provided a clue for the further studies on the underlying mechanisms coupling centrosome duplication and DNA duplication initiation during the cell cycle.

## Methods

**Cell culture and cell cycle synchronization.** HEK293, HeLa and U2OS cells were from American Type Culture Collection (ATCC) and were confirmed without mycoplasma contamination. Cells were grown at 37 °C in Dulbecco's modified Eagle medium (GIBCO) supplemented with 10% fetal bovine serum, 100 U ml$^{-1}$ penicillin and 100 μg ml$^{-1}$ streptomycin in the presence of 5% $CO_2$.

Cells synchronized at the G1/S transition were obtained via a double-thymidine-block-and-release approach. Briefly, HeLa cells were treated with 2.5 mM thymidine (Sigma) for 16 h and then released for 10 h in fresh medium. The cells were treated and blocked at the G1/S transition by a second treatment with 2.5 mM thymidine for 16 h. After the double-thymidine block, the cells were released for 3 or 6 h in fresh medium to acquire the early or late S phase cells, or for 8 or 10 h to acquire the early or late G2 phase cells.

**Plasmids and antibodies.** Human Cdc6 was cloned from a cDNA library by RT-PCR, and inserted into pEGFP-C3, pCMV-Myc, pmCherry-C2 or pGEX 4 T-1 vector. The Cdc6 WA (K208E), WB (E285G), 2A (S30A, T527A), 2D (S30D, T527D) and Cdc6 Δcy mutants were generated by PCR site-directed mutagenesis and then inserted into pEGFP-C3 or pET28a vector. GFP, mCherry and Myc tags are N-terminal to the Cdc6 WT or mutant. Human cyclin A and cyclin A 1-200 fragments were cloned from a cDNA library by RT-PCR and inserted into pEGFP-C2 or pcDNA3.1-HA vector. The cyclin A ΔCLS mutant was generated by PCR site-directed mutagenesis and then inserted into pEGFP-C2 vector. GFP-tagged cyclin A CLS and GFP-tagged cyclin A CLS DWVE-A were kindly provided by Dr Gaetan Pascreau (University of Colorado). GFP tag is N-terminal to cyclin A WT or mutant, and HA tag is C-terminal to cyclin A. Human Plk4 was cloned from a cDNA library by PCR and inserted into pEGFP-N3, pET28a or pGEX 4 T-1 vector. GFP tag is C-terminal to Plk4 WT or mutant, and GST and His tags are N-terminal to Plk4. Human Sas-6 was cloned from a cDNA library by PCR, and inserted into pEGFP-C1, pCMV-Myc, pET28a or pcDNA3.1-HA vector. GFP, Myc and His tags are N-terminal to Sas-6, and HA tag is C-terminal to Sas-6. Human STIL was cloned from a cDNA library by PCR, and inserted into pcDNA3.1-HA vector. HA tag is C-terminal to STIL.

The anti-GFP antibody used for immunoprecipitation (4 μg per sample) was generated by immunizing rabbits with bacterial expressed recombinant GFP tagged with His. The anti-Plk4 antibody used for western blotting (1:500) was generated by immunizing mice with bacterially expressed recombinant Plk4 (amino acids 1-100 at N-terminus) tagged with GST. The anti-Plk4 antibody used for immunofluorescence (1:500) was kindly provided by Dr Monica Bettencourt-Dias (Instituto Gulbenkian de Ciencia, Oeiras, Portugal). For immunofluorescence, mouse anti-Cdc6 (sc-13136, Santa Cruz Biotechnology, 1:50) and anti-Sas-6 (sc-81431, Santa Cruz, 1:100); and rabbit anti-centrin1 (12794-1-AP, Proteintech, 1:200), anti-γ-tubulin (T3559, Sigma, 1:200), anti-CP110 (sc-136629, Santa Cruz, 1:50), anti-Cep97 (A301-945A, Bethyl, 1:50), anti-STIL (ab89314, Abcam, 1:100) and anti-Cep164 (22227-1-AP, Proteintech, 1:100) antibodies were used. For immunoprecipitation, agarose beads conjugated anti-HA mouse antibody (AT0079, CMCTAG) were used. For western blotting, mouse anti-GFP (sc-9996, Santa Cruz, 1:3,000), anti-Myc (M4439, Sigma, 1:1,000) and anti-GAPDH (60004-1-Ig, Proteintech, 1:3,000); and rabbit anti-cyclin A (sc-751, Santa Cruz, 1:500), anti-HA (H6908, Sigma, 1:1,000), anti-γ-tubulin (T3559, Sigma, 1:1,000), anti-STIL (ab89314, Abcam, 1:2,000), anti-GST (sc-459, Santa Cruz, 1:1000) and anti-phosphoserine (ab9332, Abcam, 1:1,000) antibodies were used.

**HU induced centriole amplification assay.** U2OS cells were grown to 40% confluency on coverslips housed in 35-mm dishes before transfection with plasmid DNA. After transfection for 4 h, the transfected U2OS cells were treated and arrested in S phase with 16 mM hydroxyurea (HU, H8627, Sigma) for 52 h to promote centriole amplification. The cells were then stained with γ-tubulin or centrin1 antibodies for centrosome or centriole visualization, respectively, and the numbers of the centrioles and centrosomes in the transfected cells were counted.

**Plasmid DNA transfection and RNA interference.** U2OS cells at 40% confluency were transfected with plasmid DNA using Lipofectamine 2000 (Invitrogen). For each 35-mm dish, 2 μg of plasmid DNA and 5 μl of Lipofectamine 2000 were added to DMEM (final volume, 500 μl), and the mixture was incubated for 20 min. The mixture was then added to U2OS cells with 2 ml medium for 4 h, and then the medium was replaced with fresh medium.

Cdc6 siRNA (5′-AACUUCCCACCUUAUACCAGA-3′), Cdc6 siRNA-2 (5′-AA GAAUCUGCAUGUGUGAGAC-3′), Sas-6 siRNA (5′-UUAACUGUUUG GUAACUGCCCAGGG-3′), Plk4 siRNA (5′- CTCCTTTCAGACATATAAG-3′) and a non-targeting negative control siRNA (5′-UUCUCCGAACGUGUC ACGUTT-3′) were synthesized by GenePharma. U2OS cells at 30–40% confluency were transfected with Lipofectamine 2000 following the manufacturer's

instructions. For each 35-mm dish, 0.2 μM siRNA and 5 μl of Lipofectamine 2000 were added to DMEM (final volume, 500 μl), and the mixture was incubated for 20 min. The mixture was then added to cells with 1.5 ml of medium. The medium was replaced with fresh medium after 24 h. At 72 h after siRNA transfection, the cells were fixed and processed for immunofluorescence.

**Immunoprecipitation.** Approximately 2 μg of the indicated or control antibody was incubated with 25 μl of protein A conjugated-Sepharose beads (Amersham) in 500 μl of phosphate-buffer saline (PBS) for 1.5 h at 4 °C. The antibody-conjugated protein A beads were then washed with immunoprecipitation buffer (20 mM Tris, 125 mM NaCl, 0.5% NP-40, 10 mM NaF, 0.5 mM EGTA, 1 mM Na$_3$VO$_4$ and 1 mM PMSF, pH 7.5). Cells were lysed in immunoprecipitation buffer supplemented with a protease inhibitor cocktail for 15 min at 4 °C and then centrifuged at 13,680 g for 15 min. The supernatant was collected and incubated with the antibody-conjugated protein A conjugated-Sepharose beads for 2 h at 4 °C. After extensive washing, the proteins were eluted from the beads and analysed by SDS–polyacrylamide gel electrophoresis (SDS–PAGE) and western blotting.

**Western blot assays.** Protein samples were separated using SDS–PAGE and transferred to nitrocellulose filters, then blocked with 3% milk in TTBS (20 mM Tris–HCl [pH 7.4], 500 mM NaCl and 0.1% Tween 20) for 30 min, and incubated with primary antibody overnight at 4 °C. After washing three times with TTBS buffer, membrane was incubated with horseradish peroxidase-conjugated secondary antibody (Jackson, diluted 1:5,000 in 3% milk) for 1 h at room temperature and then washed with TTBS. The filter was developed for visualization by enhanced chemiluminescence and X-ray films.

All uncropped western blots and gels can be found in Supplementary Figs 8–11.

**Immunofluorescence.** For immunofluorescence, the cells were grown on coverslips, fixed in −20 °C precooled methanol for 5 min. Alternatively, in Figs 1a and 2a and Supplementary Fig. 1d, cells were permeabilized with 0.5% Triton X-100 in PBS for 40 s, fixed by 4% paraformaldehyde in PBS for 15 min and permeabilized in 0.5% Triton X-100 in PBS for 15 min. The fixed cells were incubated with primary antibodies overnight at 4 °C, washed three times with PBS and then incubated with the indicated secondary antibody for 1 h at room temperature. The coverslips were mounted with mowiol containing 1 μg ml$^{-1}$ DAPI and then examined under a DeltaVision Elite microscope. The images are the maximum intensity projections of image stacks of 1.2 μm interval from six scanning layers, with a distance between the single images of 0.2 μm. The images in Supplementary Fig. 1a,b and h are the maximum intensity projection of an image stack of 8 μm interval from eight scanning layers. The images were analysed using Velocity (ver. 6.1.1) software. Alternatively, in Figs 1a and 2a, the coverslips were examined under a Zeiss LSM 710NLO confocal microscope and projection images of confocal image stacks of 0.6 μm interval were taken. For live-cell imaging of GFP-tagged Cdc6, GFP-tagged WA and GFP-tagged WB mutants, the images are projection images of confocal image stacks of 7.5 μm interval from six scanning layers every 20 min. For live-cell image acquisition and processing, an Ultra View Vox spinning disc confocal microscope (PerkinElmer Inc.) with Velocity (ver. 6.1.1) software was used.

**Super resolution microscopy.** Three-dimensional structured illumination (3D-SIM) images were performed on an N-SIM imaging system (Nikon) equipped with a ×100/1.49 NA oil-immersion objective (Nikon) and four laser beams (405, 488,561 and 640), with a 120 nm highest resolution of captured images. Laser lines at 405, 488 and 561 nm were used for excitation. Images stacks with 0.84 μm interval were acquired and computationally reconstructed to generate super-resolution optical serial sections with two-fold extended resolution in all three axes. The reconstructed images were further processed for maximum-intensity projections and 3D-rendering with NIS-Elements AR 4.20.00 (Nikon). The image resolution measured following a Gaussian distribution with a full width at half maximum was 130 ± 3 nm in the 3D-SIM images. Centriole dots shown in these 3D-SIM images are approximately same to the highest resolution and co-localization can be concluded.

Image acquisition of STED (Stimulated Emission of Depletion microscopy) was acquired using a gated STED (gSTED) microscope (Leica TCS SP8 STED 3 ×, Leica Microsystems, Germany) equipped with a HCX PL APO ×100/1.40 NA oil objective, with a 50 nm highest resolution of captured images. The image of green channels were obtained using 488 nm excitation and 592 nm depletion. The image of red channels were acquired using 592 nm excitation and 775 nm depletion. The detection wavelength range was set to 502–568 nm for green channel and 603–700 nm for red channel. All images were acquired using the LAS AF software (Leica). The deconvolution processing was performed with Huygens Professional software (Scientific Volume Imaging, Hilversum, the Netherlands). The image resolution measured following a Gaussian distribution with a full width at half maximum was 90 ± 5 nm in our images. Centriole dots shown in these STED images are approximately same to the highest resolution and co-localization can be concluded.

**Protein expression and purification.** His-tagged Cdc6, Cdc6 2A and 2D mutants were expressed and purified from *E. coli* strain BL21 cells. His-tagged Plk4 was expressed and purified from the baculovirus/insect *Sf9* cell expression system. Briefly, bacterial pellets (induced with 0.2 mg IPTG for 14 h at 16 °C) or *Sf9* cells were lysed by sonication with extraction buffer (50 mM $NaH_2PO_4$, 300 mM NaCl and 10 mM imidazole, pH 8.0). The supernatant was then incubated with His60 Ni Superflow resin at 4 °C for 1 h. Nonspecific binding to the resin was eliminated by washing with a three-fold resin-bed-volume of washing buffer (50 mM $NaH_2PO_4$, 300 mM NaCl and 20 mM imidazole, pH 8.0). The purified proteins were then eluted from the resin with elution buffer (50 mM $NaH_2PO_4$, 300 mM NaCl and 250 mM imidazole, pH 8.0) and dialysed with PBS.

GST-tagged Plk4 and GST-tagged Cdc6 were expressed and purified from *E. coli* strain BL21 cells with PBS. Briefly, the bacterial pellet was lysed by sonication, and the supernatant was incubated with glutathione Sepharose 4B beads (GE Healthcare) at 4 °C for 1 h. Nonspecific binding to the resin was eliminated by washing with a three-fold resin-bed-volume of PBS. The purified proteins were eluted with elution buffer (50 mM Tris–HCl and 10 mM reduced glutathione, pH 8.0) and dialysed with PBS.

His-tagged Cdc6, Cdc6 2A, Cdc6 2D, Sas-6 and HA-tagged cyclin A, Sas-6, STIL proteins in Figs 1c and 6b,h,i were generated using quick coupled transcription/translation system (L1170, Promega).

**In vitro protein interaction assay.** For determination of Cdc6–Sas-6 interaction *in vitro*, His-tagged Cdc6 or its mutant protein and His-tagged Sas-6 protein were incubated with purified GST or GST-Plk4 in kinase reaction buffer (20 mM Tris–HCl, 10 mM $MgCl_2$, 25 mM KCl, 1 mM DTT, 5 μM $Na_3VO_4$ and 200 μM ATP, pH 7.5) for 30 min in 30 °C. Reactions were then supplemented with 700 μl immunoprecipitation buffer (20 mM Tris, 125 mM NaCl, 0.5% NP-40, 10 mM NaF, 0.5 mM EGTA, 1 mM $Na_3VO_4$ and 1 mM PMSF, pH 7.5) and incubated with the antibody-conjugated protein A conjugated-Sepharose beads in immunoprecipitation buffer for 2 h at 4 °C. For determination of Sas-6-STIL interaction *in vitro*, HA-tagged Sas-6 protein and HA-tagged STIL protein were incubated with purified GST or GST-Plk4 and indicated proteins in kinase reaction buffer (20 mM Tris–HCl, 10 mM $MgCl_2$, 25 mM KCl, 1 mM DTT, 5 μM $Na_3VO_4$ and 200 μM ATP, pH 7.5) for 30 min in 30 °C. Reactions were then supplemented with 700 μl immunoprecipitation buffer (20 mM Tris, 125 mM NaCl, 0.5% NP-40, 10 mM NaF, 0.5 mM EGTA, 1 mM $Na_3VO_4$ and 1 mM PMSF, pH 7.5) and incubated with the antibody-conjugated protein A conjugated-Sepharose beads for 2 h at 4 °C.

**In vitro kinase activity assays.** For *in vitro* kinase activity assay, ~2 μg of Cdc6 or its mutant protein was mixed with 1 μg of GST-tagged Plk4 in kinase buffer (20 mM Tris–HCl, 10 mM $MgCl_2$, 25 mM KCl, 1 mM DTT, 5 μM $Na_3VO_4$, 200 μM ATP and 10 μCi γ-$^{32}$P ATP, pH 7.5), and incubated at 30 °C for 30 min. The samples were then resolved by SDS–PAGE and exposed to KODAK film.

**Statistical analysis.** Approximately 100–300 cells were counted per sample, and three independent experiments were conducted. Statistical analysis was performed using GraphPad Prism software. *P* values were calculated by Student's *t*-test from the mean values of the indicated data. Significant differences were marked with asterisks (*$P < 0.05$; **$P < 0.01$; ***$P < 0.001$; N.S., no significant difference).

**Data availability.** The authors declare that all data supporting the findings of this study are available within the article and its Supplementary information files or from the corresponding author on reasonable request.

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

## Acknowledgements

We thank Drs Monica Bettencourt-Dias and Rebecca Ferguson for providing reagents and all other members of our laboratory for helpful discussions. We thank colleagues at the Core Facilities Section for assistance with 3D-SIM and STED Microscopy and BD FACSVerse, Drs Hongxia Lv for help with analysing FACS data and Chunyan Shan for help with taking STED images. This work was supported by grants from the National Natural Science Foundation of China (NSFC) (31520103906, 31371365 and 31430051) and the Ministry of Science and Technology of China (2016YFA0100501, 2016YFA0500201 and 2014CB138402).

## Author contributions

C.Z., X.X., S.H. and Q.J. designed the experiments and analysed the data. X.X. performed most of the experiments. S.H., B.Z., F.H., W.C., J.F., G.W. and S.L. performed some of the experiments. C.Z., X.X., S.H. and Q.J. wrote the manuscript.
