## [Peer Review File · Nature Communications]

Reviewers' Comments:

Reviewer #1 (Remarks to the Author)

This study by Xu and colleagues focused on the long standing question of how centriole duplication might be coupled to DNA replication in S phase. There are several links between the systems, but we are far from forming a complete picture. This topic is very exciting and is at the forefront of centrosome biology. The authors present a model where Cdc6, a component of the DNA pre-replication machinery, plays a central role in controlling centriole duplication. Specifically, the authors suggest that Cdc6 suppresses centriole duplication by binding and inhibiting the cartwheel component Sas6. To trigger duplication, Plk4 phosphorylates Cdc6 at two sites leading to Cdc6 releasing Sas6 to initial pro centriole nucleation. The experiment showing that the Cdc6-2A, but not the Cdc6-2D, mutant suppresses Plk4's driven overduplication is a particularly nice experiment that supports their model. This study is overall quite compelling and should be considered for publication if the authors can address my concerns. My major concerns revolve around data over interpretation.

Main concerns:

1) Claims of direct interactions when only IP data is provided. This is seen throughout this manuscript. IPs must not be interpreted as direct protein-protein interaction in any of the following cases

Cdc6-Aurora A, Cdc6-Plk4, Cdc6-Sas6

Aurora A interaction with Cdc6, for example, is claimed to be direct as supported by the delta201-255 truncation. This does not prove direct interaction; in fact, it could suggest that the interaction seen might be a result of pulling down the entire centrosome, or maybe just an intermediate protein. The authors should show specificity of their IPs by blotting for other centrosome component as negative controls? I suggest pulling down Cdc6 and blotting for 2 or 3 other centriole proteins not predicted to interact with Cdc6.

To truly show direct protein-protein interaction to support their claims, they must use purified proteins (which they at least have for Cdc6 and Plk4, fig 4), or at minimum a Y2H test.

Alternatively, the authors must explicitly state that they are unable to determine a direct interaction and that an intermediate(s) could exist. Cartoons of intermediates must also be indicated in their model in figure 7 so as not to suggest direct binding.

2) Related to the above, the claim that the interactions detected by IP show that they occur at the centrosome is not supported. The gradient shown in Figure 1, for example, only shows that Cdc6 is found in the centriole fraction. Also, the interaction of Plk4 with the N- and central regions of Cdc6 is claimed to be at the centrosome, but I do not understand how they can make this conclusion with the presented data.

3) The data related to Plk4, Cdc6 and Sas6 co-localization specifically at the cartwheel from SIM and STED imaging is not convincing. This type of analysis is difficult and requires much more care than what is presented in this manuscript in order to make specific claims about localization to the cartwheel. The cartoon in figure 3 is an over interpretation of the data. To make their conclusion about localization explicitly to the cartwheel they must use a total centriole marker, and relate Cdc6 localization to Sas6 in this context.

4) The use of γ -tubulin to count centriole/centrosome needs to be revised in all cases. The authors have demonstrated nicely that they can detect centrioles using Centrin as their marker figure 2A. The authors should use this same antibody to quantify all the centrosome over/under-duplication phenotypes. This includes Figure 2e and f, Figure 5a and d, Figure 6b. The centrosome field is full of cases where overexpressing a component leads to aggregation and random cytoplasmic clusters. It is important to show that they are counting centrioles.

5) The blot shown for the Cdc6-Sas6 interaction in WT and 2A, 2D mutants is unreliable for quantification. The bands are discontinuous and very weak. This must be repeated in two more

independent experiments and the Sas6 signal should be normalized to the amount of GFP-Cdc6 in the IP. Finally, the average of all three experiments should be used to calculate the ratio.

Minor Concerns

1) Can the authors comment explicitly on the timing of events in normal cells? The localization of Cdc6 on the centrosome appears to be restricted to late S or G2. This localization would explain Cdc6's role in preventing overduplication. Do the authors think that Cdc6 plays a role in suppressing duplication in G1? It would be helpful if their model depicted the normal process of Cdc6 levels at the centrosome throughout the cell cycle instead of their current model that shows under and over expression of Cdc6.

2) The authors must quantify their flow cytometry data in Figures S2 and S3 to show percent change in cell cycle populations. The conclusions drawn seem to be empirical. For example, the authors claim a G2/M arrest in the Cdc6 depletion cells on page 8, but figure S2c shows a weak arrest if any. The numbers will help support or refute their claim.

Other comments

- There is precedence for the presence of suppressors of Plk4 loss of function mutations in the *C. elegans* literature. It would be very nice to see a double knockdown of Plk4 and Cdc6. If centrioles are not there, then this supports their model, but it would be quite exciting if centrioles appear again. This could suggest Cdc6's role in linking two parallel pathways for centriole duplication.

-The authors should include reference to the Rogers and Bettencourt-Dias labs when discussing Plk4 level control in the introduction

- The sentence on page 12 starting with "After co-overexpression" was difficult to follow. Consider rewriting to state the problem first and then the experiment.

In summary, the authors must back off on their strong claims of direct binding and direct regulation, or provide stronger data to support this claim. Possible intermediates should be indicated in their final model.

Reviewer #2 (Remarks to the Author)

In this manuscript Xu et al. examine how Cdc6, a protein that is involved in regulating the initiation of DNA synthesis, also influences centriole duplication. They confirm that Cdc6 is localised to centrioles, and suggest that it is localised via an interaction with Cyclin A. They present evidence that Cdc6 acts to prevent centriole over duplication by interacting with the core centriole duplication protein Sas-6, and that this interaction can be suppressed when Plk4, another core centriole duplication protein, phosphorylates Cdc6 and so prevents its interaction with Sas-6. They conclude that Cdc6 and Plk4 normally act antagonistically to regulate centrosome duplication during the cell cycle.

There have been several previous reports that proteins involved in DNA replication, and in particular origin recognition, also appear to influence centriole duplication. The mechanism for this is unclear and this manuscript presents some interesting data that I think is very relevant and will be of sufficient general interest to warrant publication in Nature Communications. However, I have a number of concerns that should be addressed prior to publication.

1. The data from the Cyclin A/Cdc6 interaction experiments shown in Figure 1 need several

improvements and clarifications.

a) The authors show that deleting the Cyclin A interacting domain of Cdc6 blocks the recruitment of Cdc6 to centrosomes-but they don't show that this Cdc6 deletion functions normally in other ways, nor that Cyclin A still goes to centrosomes when it can't bind Cdc6. This is important, as when the authors delete the CLS in Cyclin A, the deleted protein no longer interacts with Cdc6. Thus, Cdc6 seems to be binding to the same region of CycA that is required to localise Cyclin A to centrosomes-raising the possibility that Cdc6 is actually recruiting Cyclin A to centrosomes, not the other way around. I also don't think the authors can conclude from these experiments that "Cdc6 interacts with centrosomal Cyclin A".

b) The authors go on to show that over expressing the Cyclin A CLS will block the localisation of the endogenous Cyclin A to centrosomes (although they actually don't show this, only showing that it blocks the localisation of Cdc6). This seems contradictory with the biochemical result shown in Figure 1f that Cdc6 interacts with the Cyclin A CLS (or at least the Cyclin A CLS is required for this interaction). If this is the case, one might expect that Cdc6 would be able to bind to the Cyclin A CLS that is localised at the centrosome. This apparently contradictory finding should at least be commented on.

c) The sucrose gradient data presented in Figure 2b is not very convincing, as almost every protein the authors examine appears to be very broadly spread throughout most of the gradient. In its present form this is very poor evidence and should either be dropped, or replaced with better gradients that really separate out the centrosomes from any contaminants in the preparation.

2. The experiment described Figure 2g was confusing (and not helped by the fact that the number of centrin dots was not shown at the bottom of the Figure-so I can't tell which bars represent the "normal" situation). I think the authors are claiming that these cycling cells normally have too many centrioles (3-4) and this is corrected by the over expression of Cdc6. If so, this needs to be explained better, as the Cdc6 appears to be correcting an inherent abnormality in the behaviour of these cells, so it is not correct to imply that this demonstrates a function in "normal" centriole duplication.

3. The 3D-SIM data in Figure 3 is also a bit confusing. In 3b I don't think the cartoon accurately reflects what is shown in the pictures. Cdc6 and Plk4 seem to be located in two dots close together, and they exhibit almost a reciprocal localisation (Cdc6 is highest on the dot where Plk4 is lowest). The authors also need to state clearly in the legend whether the Myc-tag is N- or C-terminal in the STED experiments.

4. The data presented in Figure 4 needs to be clarified in terms of when endogenous and over expressed proteins are being immunoprecipitated. I found it very difficult to be certain whether this data was showing interactions between the endogenous proteins, or whether every experiment relied on over expressed constructs (which I think is largely the case). If so, this should be clearly stated, and it should be highlighted that the authors have not been able to observe an interaction between the endogenous proteins (and perhaps speculate why this is the case).

5. The in vitro phosphorylation of Cdc6 by GST-Plk4 (Figure 4g) is not very convincing. I think the authors need to quantitate this data and show that in multiple experiments they observe the same pattern. It would also be important to comment on whether these phosphorylation sites are conserved in other organisms.

Reviewer #3 (Remarks to the Author)

Review of NCOMMS-16-14275 by Zhang/ CDC6 centrosomes

This paper reports some interesting data on the potential role of the DNA replication initiation protein CDC6 in controlling centriole and centrosome duplication. They show that CDC6 binds and is phosphorylated by PLK4 and that the un-phosphorylated CDC6, but not the phosphorylated CDC6 blocks centriole duplication. This suggests a model presented in Figure 7.

Much of the data in this paper supports a role for CDC6 phosphorylation in controlling centriole duplication, but there are concerns about other data that are outlined below. These concerns need addressing before this paper can be published.

Specific comments:

1. In Figure 1a, the only CDC6 protein observed is in the centrosome. Where is all the other CDC6 protein?

2. Figure 1 e. The bulk of CDC6 is clearly associated with the nucleus in this particular cell and thus the cell must be in G1 phase or early S phase. Yet in all the other panels with WT CDC6, the CDC6 not associated with the centrosomes is in the cytoplasm. So this staining cannot be compared to the WT control and the conclusion is invalid. The result could easily represent a cell cycle effect.

3. Page 7. The authors express a Cyclin A delta 201-255 and show that it does not bind to CDC6, which is fine (Figure 1 f). But then they conclude on page 7 that because CDC6 no longer associates with the centrosome, "that CDC6 specifically interacts with centrosomal Cyclin A". This is not a valid conclusion because it implies that CDC6 only interacts with Cyclin A on centrosomes. This has not been shown, despite the data in figure 1 g.

4. In Figure 1 g, expression of the Cyclin A GFP-CLS prevented CDC6 from localizing to centrosomes. If CDC6 is not there, why did this not induce centrosome over-duplication? There is no evidence for this in Figure 1 g since gamma tubulin staining is normal. This would be expected based on the conclusions of this paper.

5. Supplemental Figure 2 c. There appears to be a significant difference between the two CDC6 siRNAs, with siCdc6 producing more 4C cells than CDC6-2 siRNA. This is not explained.

6. Supplemental Figure 2 c. The authors label the DNA content as 2N and 4N. This is incorrect. N refers to the number of chromosomes. DNA content is measured in C. Thus it should be 2C and 4C.

7. Figure 2 g. There are multiple pairs of bars in this figure but the legend does not explain what they are. The different bars of GFP versus GFP-CDC6 show opposite results. Is the figure missing labels on the X axis? What is the N.S. result? It is not mentioned on page 9.

8. In contrast to the description on page 9, CDC6 does not localize with PLK4 in figure 3 b, but to one of the PLK4 spots. Also, CDC6 localizes near, but not overlapping with SAS6. The summary diagram is misleading.

9. Why is the CDC6 staining different in Figure 3 b versus 3 c? (comparing upper panel inserts in each figure)

10. To claim that CDC6 localizes on the "cartwheel" (bottom of page 9 and Figure 3 legend and abstract) based on the data presented is an over-interpretation.

11. Figure 4 C. The GFP-PLK4 and GFP-K41M mutant lanes are different. The GFP-PLK4 is missing in the experiment in the GFP IP. A prominent band is present in the GFP-P41M lane 3, but no corresponding band is present in lane 2. There is no explanation for this. Furthermore, how is CDC6 pulled down with anti-GFP antibodies if GFP-PLK4 is missing?

12. Figure 4 g. It would be appropriate to determine if CDC^Δ is phosphorylated on these residues in cells.

13. Figure 5 b and c. These results could be explained by a general inhibition of cell cycle progression from G1 phase to S phase and beyond by CDC6 or the CDC6-2A mutant, but not the CDC^Δ-2D mutant. It is necessary to perform flow cytometry for DNA content in this experiment.

14. Figure 6 d. The differences described on page 13 as the CDC6-2A binding "stronger" are not convincing.

15. Figure 6 e. The blot of the levels of PLK4 shows little difference in PLK4 levels and thus the interpretation of this experiment is not justified.

16. Page 13. This is an inappropriate conclusion here. Just because knock down of SAS-6 blocks centriole duplication in the absence of CDC6 does not mean, one way or the other, that CDC6 directly blocks SAS-6. It means that SAS-6 is required for centriole duplication. The same might be true for HU induced amplification and one would not conclude that SAS-6 is regulated by HU!

17. Page 14 and abstract. The authors conclude that CDC6 binds SAS-6, which is fine, but they also conclude that it inhibits SAS-6 activity. Can they point to the data that supports this conclusion? (note point above). The only data that is relevant here is Figure 6 f, but I note that overexpression of Myc-SAS-6 in the top panel yields a large SAS-6 cluster, and indeed CDC6 blocks this, but the CDC6-2D mutant does not yield this large cluster, only extra centriole pairs. This is not the same as phosphorylation of CDC6 by PLK4 regulating SAS-6.

Point-by-point answers

Reviewers' comments:

Reviewer #1 (Remarks to the Author):

This study by Xu and colleagues focused on the long standing question of how centriole duplication might be coupled to DNA replication in S phase. There are several links between the systems, but we are far from forming a complete picture. This topic is very exciting and is at the forefront of centrosome biology. The authors present a model where Cdc6, a component of the DNA pre-replication machinery, plays a central role in controlling centriole duplication. Specifically, the authors suggest that Cdc6 suppresses centriole duplication by binding and inhibiting the cartwheel component Sas6. To trigger duplication, Plk4 phosphorylates Cdc6 at two sites leading to Cdc6 releasing Sas6 to initial pro centriole nucleation. The experiment showing that the Cdc6-2A, but not the Cdc6-2D, mutant suppresses Plk4's driven overduplication is a particularly nice experiment that supports their model. This study is overall quite compelling and should be considered for publication if the authors can address my concerns. My major concerns revolve around data over interpretation.

Main concerns:

Comment 1) Claims of direct interactions when only IP data is provided. This is seen throughout this manuscript. IPs must not be interpreted as direct protein-protein interaction in any of the following cases

Cdc6-Aurora A, Cdc6-Plk4, Cdc6-Sas6

Aurora A interaction with Cdc6, for example, is claimed to be direct as supported by the delta201-255 truncation. This does not prove direct interaction; in fact, it could suggest that the interaction seen might be a result of pulling down the entire centrosome, or maybe just an intermediate protein. The authors should show specificity of their IPs by blotting for other centrosome component as negative controls? I suggest pulling down Cdc6 and blotting for 2 or 3 other centriole proteins not predicted to interact with Cdc6.

To truly show direct protein-protein interaction to support their claims, they must use purified proteins (which they at least have for Cdc6 and Plk4, fig 4), or at minimum a Y2H test. Alternatively, the authors must explicitly state that they are unable to determine a direct interaction and that an intermediate(s) could exist. Cartoons of intermediates must also be indicated in their model in figure 7 so as not to suggest direct binding.

Answer 1: We are sorry about the inappropriate description of immunoprecipitation. We have now removed the “direct protein-protein interaction” when we describe the immunoprecipitation experiments.

It is a very good suggestion of showing specificity of IP by blotting for other centrosome component as negative controls. Now we have performed IP by Cdc6 antibody, blotted for three centriole proteins Cep164, CPAP and STIL, and found that there are no interactions with Cdc6, excluding the possibility that Cdc6 interacts with Sas-6, Plk4, or cyclin A simply by pulling down the entire centrosome (new Fig. 6a).

Following the reviewer's suggestion, we have now included the *in vitro* protein-protein interaction of Cdc6-Plk4 (original Supplementary Fig. 4a moved to new Fig. 4b); Cdc6-cyclin A (new Fig. 1c); Cdc6-Sas-6 (new Fig. 6b), to show direct protein-protein interactions.

Comment 2) Related to the above, the claim that the interactions detected by IP show that they occur at the centrosome is not supported. The gradient shown in Figure 1, for example, only shows that Cdc6 is found in the centriole fraction. Also, the interaction of Plk4 with the N- and central regions of Cdc6 is claimed to be at the centrosome, but I do not understand how they can make this conclusion with the presented data.

Answer 2: We are sorry about this misleading description. We have removed “the interaction at the centrosome” from the conclusion throughout the manuscript.

Comment 3) The data related to Plk4, Cdc6 and Sas6 co-localization specifically at the cartwheel from SIM and STED imaging is not convincing. This type of analysis is difficult and requires much more care than what is presented in this manuscript in order to make specific claims about localization to the cartwheel. The cartoon in figure 3 is an over interpretation of the data. To make their conclusion about localization explicitly to the cartwheel they must use a total centriole marker, and relate Cdc6 localization to Sas6 in this context.

Answer 3: We have repeated the SIM and STED imaging in Figure 3 and included the co-localization of Cdc6 and Sas-6 or Plk4 in the context of a distal side-localized centriole marker CP110 (new Fig. 3b,d). To confirm the co-localization of Cdc6 with cartwheel components, we have now also included the co-staining of Cdc6 and Plk4, Cdc6 and Sas-6, Cdc6 and STIL under STED (new Fig. 3c,d). The results showed that Cdc6 colocalizes with Plk4 and cartwheel proteins Sas-6 and STIL, but not the distal side-localized centriole protein CP110. Also, we replaced the previous conclusion that “Cdc6 colocalizes with Plk4 and Sas-6 on the cartwheel” with “Cdc6 colocalizes with Plk4, and cartwheel proteins Sas-6 and STIL”.

Comment 4) The use of g-tubulin to count centriole/centrosome needs to be revised in all cases. The authors have demonstrated nicely that they can detect centrioles using Centrin as their marker figure 2A. The authors should use this same antibody to quantify all the centrosome over/under-duplication phenotypes. This includes Figure 2e and f, Figure 5a and d, Figure 6b. The centrosome field is full of cases where overexpressing a component leads to aggregation and random cytoplasmic clusters. It is important to show that they are counting centrioles.

Answer 4: This is a very good suggestion. We have now included centrin1 staining as the centriole marker in new Fig. 2e,f; new Fig. 5a,b,d and e; and new Fig. 6d,e, and quantified centrosome over-duplication by counting centrioles. The original Fig. 2e,f are now Supplementary Fig. 3c,d; The original Fig. 5a,b are now Supplementary Fig. 5a,b; The original Fig. 6b,c are now Supplementary Fig. 6c,d.

Comment 5) The blot shown for the Cdc6-Sas6 interaction in WT and 2A, 2D mutants is unreliable for quantification. The bands are discontinuous and very weak. This must be repeated in two more independent experiments and the Sas6 signal should be normalized to the amount of GFP-Cdc6 in the IP. Finally, the average of all three experiments should be used to calculate the ratio.

Answer 5: Thank you for pointing out this confusion. We have now quantified the binding between Sas-6 and Cdc6 WT, 2A, or 2D from three independent immunoprecipitation experiments (new Supplementary Fig. 6g,h) and included the representative result and quantitation in new Fig. 6g. We have now also included the in vitro interaction between Cdc6 WT, 2A, or 2D with Sas-6 in new Fig. 6h. The in vitro results confirmed that Cdc6 2A mutant binds to Sas-6 stronger than Cdc6 2D mutant or phosphorylated Cdc6 WT incubated with Plk4.

Minor Concerns

Comment 1) Can the authors comment explicitly on the timing of events in normal cells? The localization of Cdc6 on the centrosome appears to be restricted to late S or G2. This localization would explain Cdc6's role in preventing overduplication. Do the authors think that Cdc6 plays a role in suppressing duplication in G1? It would be helpful if their model depicted the normal process of Cdc6 levels at the centrosome throughout the cell cycle instead of their current model that shows under and over expression of Cdc6.

Answer 1: This is a good point. We have now included a video showing Cdc6 localization from G2 to next G1/S phase in new Supplementary Fig. 1f and new Supplementary Movie 4, showing that Cdc6 localizes in nucleus in G1 phase, rather than centrosome. We also added the comment "The fact that Cdc6 localizes only in S and G2 phase, but not in G1 phase, suggests that the centrosome localized Cdc6 functions only in S and G2 phases." in the results.

We also modified the model by first depicting the normal cell cycle dependent function of Cdc6 in centrosome duplication, and then discussing the abnormal expression of Cdc6 caused centrosome duplication defects.

Comment 2) The authors must quantify their flow cytometry data in Figures S2 and S3 to show percent change in cell cycle populations. The conclusions drawn seem to be empirical. For example, the authors claim a G2/M arrest in the Cdc6 depletion cells on page 8, but figure S2c shows a weak arrest if any. The numbers will help support or refute their claim.

Answer 2: We thank this reviewer for this good suggestion. We have now included quantifications of the flow cytometry data from the original Supplementary Fig. 2c (new Supplementary Fig. 2c,d) and the original Supplementary Fig. 3c,d (new Supplementary Fig. 5f,g).

About the G2/M arrest percentage in Supplementary Fig. 2c, Cdc6 siRNA-2 showed weaker G2/M phase arrest compared to Cdc6 siRNA. It is because Cdc6 siRNA-2 depletion efficiency is lower than Cdc6 siRNA. We have repeated this experiments three times, quantified the decrease of G1 phase cells and increase of G2/M phase cells, and showed that although with a lower efficiency of depletion by Cdc6 siRNA-2, the defects are significant compared to control (new Supplementary Fig. 2c,d). Because of the low efficiency of siRNA-2, we used Cdc6 siRNA for later experiments.

Other comments

Comment - There is precedence for the presence of suppressors of Plk4 loss of function mutations in the *c. elegans* literature. It would be very nice to see a double knockdown of Plk4 and Cdc6. If centrioles are not there, then this supports their model, but it would be quite

exciting if centrioles appear again. This could suggest Cdc6's role in linking two parallel pathways for centriole duplication.

Answer: We have performed the double knockdown of Plk4 and Cdc6, and included the results in new Supplementary Fig. 5c,d. The result showed that simultaneous depletion of Cdc6 and Plk4 restored the centriole numbers which was decreased after Plk4 depletion, indicating that Cdc6 is not a Plk4 inhibitor, and that Cdc6 inhibits centrosome duplication downstream of Plk4, or in a parallel pathway from Plk4.

Comment -The authors should include reference to the Rogers and Bettencourt-Dias labs when discussing Plk4 level control in the introduction

Answer: We have now added in the introduction "Plk4 protein level is strictly controlled by SCF/SLimb ubiquitin ligase mediated degradation to block centriole reduplication" followed by these two references to Bettencourt-Dias: *Curr Biol.* 2009 Jan 13;19(1):43-9. (The SCF/Slimb Ubiquitin Ligase Limits Centrosome Amplification through Degradation of SAK/PLK4), and to Rogers: *J Cell Biol.* 2009 Jan 26;184(2):225-39. (The SCF/Slimb ubiquitin ligase regulates Plk4/Sak levels to block centriole reduplication).

Comment - The sentence on page 12 starting with "After co-overexpression" was difficult to follow. Consider rewriting to state the problem first and then the experiment.

Answer: We have now rewritten all the descriptions with stating the questions first followed by the experimental methods throughout the manuscript.

Comment: In summary, the authors must back off on their strong claims of direct binding and direct regulation, or provide stronger data to support this claim. Possible intermediates should be indicated in their final model.

Answer: We thank this reviewer for all the suggestions and comments, which greatly helped us to improve our manuscript. To address the concerns, we have now showed the direct binding between Cdc6-Plk4 (original Supplementary Fig. 4a moved to new Fig. 4b); Cdc6-cyclin A (new Fig. 1c); Cdc6-Sas-6 (new Fig. 6b). We also provided strong evidence of direct regulation both in vitro and in vivo that Cdc6 phosphorylation by Plk4 regulates the interaction between Cdc6 and Sas-6 (new Fig. 6h). We further identified that the phosphorylation of Cdc6 by Plk4 regulates the interaction between Sas-6 and its cartwheel partner protein STIL. This has greatly improved our understanding of how Cdc6 inhibits centrosome duplication through inhibiting Sas-6.

Reviewer #2

In this manuscript Xu et al. examine how Cdc6, a protein that is involved in regulating the initiation of DNA synthesis, also influences centriole duplication. They confirm that Cdc6 is localised to centrioles, and suggest that it is localised via an interaction with Cyclin A. They present evidence that Cdc6 acts to prevent centriole over duplication by interacting with the core centriole duplication protein Sas-6, and that this interaction can be suppressed when Plk4, another core centriole duplication protein, phosphorylates Cdc6 and so prevents its interaction with Sas-6. They conclude that Cdc6 and Plk4 normally act antagonistically to regulate centrosome duplication during the cell cycle.

There have been several previous reports that proteins involved in DNA replication, and in particular origin recognition, also appear to influence centriole duplication. The mechanism for this is unclear and this manuscript presents some interesting data that I think is very relevant and will be of sufficient general interest to warrant publication in Nature Communications. However, I have a number of concerns that should be addressed prior to publication.

Comment 1. The data from the Cyclin A/Cdc6 interaction experiments shown in Figure 1 need several improvements and clarifications.

Comment a) The authors show that deleting the Cyclin A interacting domain of Cdc6 blocks the recruitment of Cdc6 to centrosomes-but they don't show that this Cdc6 deletion functions normally in other ways, nor that Cyclin A still goes to centrosomes when it can't bind Cdc6. This is important, as when the authors delete the CLS in Cyclin A, the deleted protein no longer interacts with Cdc6. Thus, Cdc6 seems to be binding to the same region of CycA that is required to localise Cyclin A to centrosomes-raising the possibility that Cdc6 is actually recruiting Cyclin A to centrosomes, not the other way around. I also don't think the authors can conclude from these experiments that "Cdc6 interacts with centrosomal Cyclin A".

Answer 1a: Thank you for pointing out this confusion. We have now assessed whether Cdc6 recruits cyclin A to centrosomes and included the result in new Supplementary Fig. 1h. The results showed that Cdc6 depletion doesn't affect the centrosomal localization of cyclin A, suggesting that cyclin A localizes to centrosomes when it can not bind Cdc6, and cyclin A is not recruited to centrosomes by Cdc6.

Cyclin A domain depletion Cdc6 (denoted as Cdc6 Δ cy) was reported to localize in the nucleus with normal functions in DNA replication (EMBO J. 1999 Jan 15;18(2):396-410.). Therefore, it is unlikely that Cdc6 Δ cy has misfolding problem which causes its incapability of centrosome localization.

We are sorry about the misleading conclusion "Cdc6 interacts with centrosomal cyclin A". We have now changed it to "the interaction between Cdc6 and cyclin A requires CLS domain in cyclin A."

Comment b) The authors go on to show that over expressing the Cyclin A CLS will block the localisation of the endogenous Cyclin A to centrosomes (although they actually don't show this, only showing that it blocks the localisation of Cdc6). This seems contradictory with the biochemical result shown in Figure 1f that Cdc6 interacts with the Cyclin A CLS (or at least the Cyclin A CLS is required for this interaction). If this is the case, one might expect that Cdc6 would be able to bind to the Cyclin A CLS that is localised at the centrosome. This apparently contradictory finding should at least be commented on.

Answer 1b: This is a good point. We have now included the endogenous cyclin A localization after cyclin A CLS overexpression (new Supplementary Fig. 1i). The results showed that endogenous cyclin A centrosome localization is abolished in cyclin A CLS WT but not CLS DWVE-A mutant transfected cells.

We also performed IP experiment to test the interaction between Cdc6 and cyclin A WT, CLS, or CLS depletion truncation (denoted as Δ CLS). The results showed that both CLS and Δ CLS truncates of cyclin A interact with Cdc6 much more weakly compared to the full length WT cyclin A (new Fig. 1e). This result suggests that cyclin A CLS is not sufficient to target Cdc6 to the centrosomes, indicating that there might be another domain that is required for its binding to Cdc6. At the same time, the substantial loss of endogenous cyclin A from centrosomes after

cyclin A CLS overexpression causes the dislocation of Cdc6 from centrosomes. We also included this comment in the results.

It was also reported that CLS domain of cyclin E and cyclin A could both abolish the centrosome localization of Mcm5 and Orc1 by replacing the endogenous cyclin E or cyclin A at the centrosome (J Cell Sci. 2008 Oct 1;121(Pt 19):3224-32; J Cell Sci. 2010 Aug 15;123(Pt 16):2743-9). Similar to Cdc6, both Mcm5 and Orc1 interact with CLS, but to a much weaker extent when compared to the full length cyclin A. The same reason as we described above for Cdc6 might be applied to Mcm5 and Orc1.

Comment c) The sucrose gradient data presented in Figure 2b is not very convincing, as almost every protein the authors examine appears to be very broadly spread throughout most of the gradient. In its present form this is very poor evidence and should either be dropped, or replaced with better gradients that really separate out the centrosomes from any contaminants in the preparation.

Answer 1c: This is a good suggestion. We agree with the reviewer that the original Figure 2b sucrose gradient data did not add much to the evidence that Cdc6 localizes at the centrosome as we already showed that both endogenous and exogenous Cdc6 localize at the centrosome by immunofluorescence microscopy. Following this reviewer's suggestion, we depleted the sucrose gradient data from the current manuscript.

Comment 2. The experiment described Figure 2g was confusing (and not helped by the fact that the number of centriole dots was not shown at the bottom of the Figure-so I can't tell which bars represent the "normal" situation). I think the authors are claiming that these cycling cells normally have too many centrioles (3-4) and this is corrected by the over expression of Cdc6. If so, this needs to be explained better, as the Cdc6 appears to be correcting an inherent abnormality in the behaviour of these cells, so it is not correct to imply that this demonstrates a function in "normal" centriole duplication.

Answer 2: We are sorry that the missing labels at the bottom of this figure in the previous version. We have now added the centriole numbers as "2", "3-4" or ">4" on the X axis in Fig. 2g. This result showed that a significant increase of the cells with 2 centrioles and a significant reduction of cells with 3 to 4 centrioles in the Cdc6 transfected cells than the GFP vector transfected cells, indicating that overexpressing Cdc6 restrained the centrosome duplication in normal cell cycle during 50 h observation time. We believe Cdc6 inhibits centriole duplication not only in the abnormal conditions such as HU treatment, but also in normal cell cycle. The expression and activity of both Plk4 and Cdc6 must be tightly balanced to prevent over-duplication of the centrosome to ensure normal centrosome duplication during the cell cycle. When Cdc6 is overexpressed for long time (50 h) (Fig. 2g and Fig. 5c) or Plk4 is depleted (new Supplemental Fig. S5c), centriole duplication is inhibited. When Cdc6 is depleted (Fig. 2a and Supplemental Fig. 3a) or Plk4 is overexpressed (Fig. 5d), centriole is over-duplicated. When Cdc6 and Plk4 are co-overexpressed (Fig. 5d) or depleted simultaneously (new Supplemental Fig. 5c), the opposing activities of Cdc6 and Plk4 ensure normal centriole duplication.

Comment 3. The 3D-SIM data in Figure 3 is also a bit confusing. In 3b I don't think the cartoon accurately reflects what is shown in the pictures. Cdc6 and Plk4 seem to be located in two dots close together, and they exhibit almost a reciprocal localisation (Cdc6 is highest on the dot where Plk4 is lowest). The authors also need to state clearly in the legend whether the Myc-tag is N- or C-terminal in the STED experiments.

Answer 3: Reviewer 1 has similar concerns about Figure 3. We have repeated the SIM and STED imaging in Figure 3, and included the co-localization of Cdc6 and Sas6 or Plk4 in the context of a centriole marker CP110 (new Fig. 3b,d). To confirm the co-localization of Cdc6

with cartwheel components, we have now also included the co-staining of Cdc6 and Plk4, Cdc6 and Sas-6, Cdc6 and STIL under STED (new Fig. 3c,d). The results showed that Cdc6 colocalizes with Plk4 and cartwheel proteins Sas-6 and STIL, but not the distal side-localized centriole protein CP110.

We have added in the methods that GFP-tag is N-terminal to Sas-6, and GFP-tag is C-terminal to Plk4 in the STED experiments. We also added in the methods the tag information of all the plasmids we used through the manuscript. To make it clear, we also re-labelled in the figures and text as “tag - protein” to show tag is N-terminal to the protein, and “protein - tag” to show tag is C-terminal to the protein.

Comment 4. The data presented in Figure 4 needs to be clarified in terms of when endogenous and over expressed proteins are being immunoprecipitated. I found it very difficult to be certain whether this data was showing interactions between the endogenous proteins, or whether every experiment relied on over expressed constructs (which I think is largely the case). If so, this should be clearly stated, and it should be highlighted that the authors have not been able to observe an interaction between the endogenous proteins (and perhaps speculate why this is the case).

Answer 4: We are sorry about this confusion. We have now added clear description as “the interaction between endogenous Cdc6 and Plk4 in vivo by co-IP with Cdc6 antibody (Fig. 4a)” or “the interaction between overexpressed Cdc6 and Plk4 was also confirmed by reciprocal co-IP using GFP-Cdc6 or GFP-Plk4 as the bait (Fig. 4c; Supplementary Fig. 4a)” in the results. For endogenous proteins interaction, we detected the endogenous interaction of Cdc6 and Plk4 by Cdc6 antibody immunoprecipitation in Fig. 4a, and the endogenous interaction of Cdc6 and Sas-6 by Cdc6 antibody immunoprecipitation in Fig. 6a.

Comment 5. The in vitro phosphorylation of Cdc6 by GST-Plk4 (Figure 4g) is not very convincing. I think the authors need to quantitate this data and show that in multiple experiments they observe the same pattern. It would also be important to comment on whether these phosphorylation sites are conserved in other organisms.

Answer 5: We thank this reviewer for this good suggestion. We have repeated the in vitro phosphorylation of Cdc6 by Plk4 in two additional independent experiments (Supplementary new Fig. 4f,g). Also, we have now quantitated the phosphorylation of Cdc6 WT and mutants normalizing to the intensity of the protein levels from these three independent experiments (Fig. 4h).

We compared S30 and T527 on Cdc6 in human, mouse, drosophila, rat, xenopus, zebrafish, and yeast. Unfortunately, they are not conserved in these organisms. However, S30 and T527 are conserved between human (*Homo sapiens*; Uniprot: Q99741) and monkey (*Chlorocebus sabaues* / *Green monkey*; Uniprot: A0A0D9S323). We are speculating that the phosphorylation sites of Cdc6 by Plk4 are not conserved in lower organisms. First, Cdc6 behaves differently in yeast and human in terms of DNA replication initiation and preventing DNA replication re-initiation (Genes Dev. 1997 Nov 1;11(21):2767-79; Curr Biol. 2000 Mar 9;10(5):231-40; EMBO J. 1999 Jan 15;18(2):396-410.). The role of Cdc6 in preventing centrosome overduplication might also be different among organisms. Second, the other two DNA replication initiation proteins Orc1, Mcm5 functioning in centrosome duplication was only reported in human cells but not in other organisms (J Cell Sci. 2008 Oct 1;121(Pt 19):3224-32; J Cell Sci. 2010 Aug 15;123(Pt 16):2743-9; Science. 2009 Feb 6;323(5915):789-93.). Therefore, it is unknown whether these two DNA replication initiation proteins' roles in centrosome duplication are conserved across organisms. Third, centriole duplication mechanism is not all conserved through organisms. For example, Sas-6 is only phosphorylated in *C. elegans* by Plk4 but not in *Drosophila* or human (Dev Cell. 2009 Dec;17(6):900-7; Curr Biol. 2014 Nov

3;24(21):2526-32.). In *C. elegans*, Sas-6 interacts directly with SAS5/Ana2 (Nat Cell Biol. 2005 Feb;7(2):115-25.). However, in human, Sas-6 interacts with STIL (the human counterpart of SAS5/Ana2) in a Plk4 phosphorylation depend manner (J Cell Biol. 2015 Jun 22;209(6):863-78). Therefore, it might be possible that Cdc6 conveys the inhibition of centrosome over-duplication by interaction with Sas-6 through different regulatory mechanisms other than Plk4 phosphorylation in lower organisms.

Reviewer #3

Review of NCOMMS-16-14275 by Zhang/ CDC6 centrosomes

This paper reports some interesting data on the potential role of the DNA replication initiation protein CDC6 in controlling centriole and centrosome duplication. They show that CDC6 binds and is phosphorylated by PLK4 and that the un-phosphorylated CDC6, but not the phosphorylated CDC6 blocks centriole duplication. This suggests a model presented in Figure 7.

Much of the data in this paper supports a role for CDC6 phosphorylation in controlling centriole duplication, but there are concerns about other data that are outlined below. These concerns need addressing before this paper can be published.

Specific comments:

Comment 1. In Figure 1a, the only CDC6 protein observed is in the centrosome. Where is all the other CDC6 protein?

Answer 1: We are sorry about this confusion. Cdc6 localizes in nucleus in G1 phase, and in cytoplasm in S and G2 phases. Centrosomal Cdc6 localization is only observed in S and G2 phases. The solo centrosome localization of Cdc6 in S and G2 phase shown in Fig. 1a is because that the image stack (0.6 μm interval) of this image only contains centrosome under the confocal microscopy, however, the cytoplasm fraction of Cdc6 are out of the scanning focus. We have now added the image stack interval of different images in the microscopy method and the figure legends.

We have also included a 8 μm interval image stack of Cdc6 staining in new Supplementary Fig. 1a, showing the concurrent localization of Cdc6 in both cytoplasm and centrosome in S and G2 phase.

Comment 2. Figure 1 e. The bulk of CDC6 is clearly associated with the nucleus in this particular cell and thus the cell must be in G1 phase or early S phase. Yet in all the other panels with WT CDC6, the CDC6 not associated with the centrosomes is in the cytoplasm. So this staining cannot be compared to the WT control and the conclusion is invalid. The result could easily represent a cell cycle effect.

Answer 2: We are sorry about this confusion. Cdc6 Δcy was reported to be restricted in the nucleus and can not be exported to cytoplasm because of its incapability of binding cyclin A, and DNA replication is not prevented in Cdc6 Δcy transfected cells, thus the cell cycle is not changed by transfecting Cdc6 Δcy (EMBO J. 1999 Jan 15;18(2):396-410.). In this case, we are

testing whether Cdc6 Δ cy is defective in centrosome localization. We have now added this description in the results.

Comment 3. Page 7. The authors express a Cyclin A delta 201-255 and show that it does not bind to CDC6, which is fine (Figure 1 f). But then they conclude on page 7 that because CDC6 no longer associates with the centrosome, “that CDC6 specifically interacts with centrosomal Cyclin A”. This is not a valid conclusion because it implies that CDC6 only interacts with Cyclin A on centrosomes. This has not been shown, despite the data in figure 1 g.

Answer 3: We are sorry about the misleading conclusion. We have now changed “that CDC6 specifically interacts with centrosomal cyclin A” to “the interaction between Cdc6 and cyclin A requires CLS domain in cyclin A”.

Comment 4. In Figure 1 g, expression of the Cyclin A GFP-CLS prevented CDC6 from localizing to centrosomes. If CDC6 is not there, why did this not induce centrosome over-duplication? There is no evidence for this in Figure 1 g since gamma tubulin staining is normal. This would be expected based on the conclusions of this paper.

Answer 4: This is a good point. We expressed GFP-cyclin A CLS for 14 h (less than 24 h) in this experiment. However, the over-duplication of centrosome can only be observed in more than one cell cycle. Centrosome over-duplication was observed 72 h after Cdc6 siRNA transfection. However, prolonged overexpression of GFP-cyclin A CLS leads to cell death. It is reported that the CLS overexpression for a long time inhibits DNA replication, resulting in defects of cell division (Proc Natl Acad Sci. 2010 Feb 16;107(7):2932-7), and we verified this result according to our detection. We also communicated with Dr. Rebecca Ferguson who gave us the plasmid as a gift, she had reminded us about the toxicity of this plasmid when sending us the plasmid. Therefore, we cannot verify the centrosome over-duplication caused by missing Cdc6 at centrosomes by overexpressing GFP-cyclin A CLS.

Comment 5. Supplemental Figure 2 c. There appears to be a significant difference between the two CDC6 siRNAs, with siCdc6 producing more 4C cells than CDC6-2 siRNA. This is not explained.

Answer 5: Reviewer 1 has the same concern. About the G2/M arrest percentage in Supplementary Fig. 2c, Cdc6 siRNA-2 showed weaker G2/M arrest compared to Cdc6 siRNA-1. It is because Cdc6 siRNA-2 depletion efficiency is lower than Cdc6 siRNA-1. We have repeated this experiment three times, quantified the decrease of G1 phase cells and increase of G2/M phase cells, and showed that although with a lower efficiency of depletion by Cdc6 siRNA-2, the defects are significant compared to control (new Supplementary Fig. 2c,d). Because of the low efficiency of siRNA-2, we used Cdc6 siRNA for later experiments.

Comment 6. Supplemental Figure 2 c. The authors label the DNA content as 2N and 4N. This is incorrect. N refers to the number of chromosomes. DNA content is measured in C. Thus it should be 2C and 4C.

Answer 6: We have now corrected to “2C and 4C” in Supplementary Fig. 2c and throughout all the other figures.

Comment 7. Figure 2 g. There are multiple pairs of bars in this figure but the legend does not explain what they are. The different bars of GFP versus GFP-CDC6 show opposite results. Is the figure missing labels on the X axis? What is the N.S. result? It is not mentioned on page 9.

Answer 7: We are sorry that the missing labels at the bottom of the figure in the old version. We have now added the centriole numbers as “2”, “3-4” or “>4” on the X axis in Fig. 2g. This

result showed that a significant increase of the cells with 2 centrioles and a significant reduction of cells with 3 to 4 centrioles in the Cdc6 transfected cells than the GFP vector transfected cells, indicating that overexpressing Cdc6 restrained the centrosome duplication in normal cell cycle during 50 h observation time. N.S result shows the percentage of abnormal centriole amplification with more than 4 centrin dots. We also added this description in the results.

Comment 8. In contrast to the description on page 9, CDC6 does not localize with PLK4 in figure 3 b, but to one of the PLK4 spots. Also, CDC6 localizes near, but not overlapping with SAS6. The summary diagram is misleading.

Answer 8: We have repeated the SIM imaging of the endogenous Cdc6 and GFP-tagged Plk4 or GFP-tagged Sas6 in new Fig. 3b. The result showed that Cdc6 co-localizes with Plk4 and Sas-6 but not with a distal side-localized centriole marker CP110. We have also included the co-staining of Cdc6 and Plk4, Cdc6 and Sas-6 under STED (new Fig. 3c,d).

Comment 9. Why is the CDC6 staining different in Figure 3 b versus 3 c? (comparing upper panel inserts in each figure)

Answer 9: We have repeated the SIM and STED imaging of Cdc6 and Plk4 or Sas-6 in Fig. 3b,c. The SIM imaging in new Fig. 3b and the STED imaging in new Fig. 3c both showed that Cdc6 colocalizes with Plk4 and Sas-6.

Comment 10. To claim that CDC6 localizes on the “cartwheel” (bottom of page 9 and Figure 3 legend and abstract) based on the data presented is an over-interpretation.

Answer 10: To confirm the co-localization of Cdc6 with cartwheel components, we have now included the co-staining of Cdc6 and Sas-6 or STIL under STED (New Fig. 3c,d). The results showed that Cdc6 colocalizes with cartwheel components Sas-6 and STIL. Also, we replaced the previous conclusion that “Cdc6 colocalizes with Plk4 and Sas-6 on the cartwheel” with “Cdc6 colocalizes with Plk4, and cartwheel proteins Sas-6 and STIL”.

Comment 11. Figure 4 C. The GFP-PLK4 and GFP-K41M mutant lanes are different. The GFP-PLK4 is missing in the experiment in the GFP IP. A prominent band is present in the GFP-P41M lane 3, but no corresponding band is present in lane 2. There is no explanation for this. Furthermore, how is CDC6 pulled down with anti-GFP antibodies if GFP-PLK4 is missing?

Answer 11: Thank you for pointing out this mistake. We have repeated the immunoprecipitation of GFP-Plk4 in Fig. 4c. The result showed that with similarly immunoprecipitated GFP-Plk4 WT and K41M mutant, Cdc6 interacts with WT Plk4 but not K41M mutant.

Comment 12. Figure 4 g. It would be appropriate to determine if CDC6 is phosphorylated on these residues in cells.

Answer 12: This is a very good suggestion. We have now detected the phosphorylation of Cdc6 WT, S30A, T527A, and 2A mutant on a Phos-Tag™ (Wako) acrylamide to show the phosphorylation of Cdc6 in cells (new Fig. 4f). The results showed that Cdc6 WT is phosphorylated in cells. Cdc6 S30A or T527A mutant partially decreased Cdc6 phosphorylation compared to Cdc6 WT, and the Cdc6 2A double alanine mutant further decreased Cdc6 phosphorylation (new Fig. 4f). This result suggests that Cdc6 is phosphorylated on S30 and T527 in cells.

Comment 13. Figure 5 b and c. These results could be explained by a general inhibition of cell cycle progression from G1 phase to S phase and beyond by CDC6 or the CDC6-2A mutant, but not the CDC6-2D mutant. It is necessary to perform flow cytometry for DNA content in this experiment.

Answer 13: We thank this reviewer for this good suggestion. We have now included the flow cytometry analyses of Cdc6 WT, 2A, or 2D mutant transfected cells after HU treatment (new Supplementary Fig. 5e) or in normal cell cycle (new Supplementary Fig. 5f).

Comment 14. Figure 6 d. The differences described on page 13 as the CDC6-2A binding “stronger” are not convincing.

Answer 14: Reviewer 1 has the same concern. We have now quantified the binding between Sas6 and Cdc6 WT, 2A, 2D from three independent immunoprecipitation experiments (new Supplementary Fig. 6g,h) and included the representative result and quantitation in new Fig. 6g. The result showed that Cdc6 2A mutant binds to Sas6 stronger than Cdc6 WT or Cdc6 2D mutant. We have also included the in vitro interaction between Cdc6 WT, 2A or 2D with Sas-6 in new Fig. 6h. The in vitro results confirmed that Cdc6 2A mutant binds to Sas-6 stronger than Cdc6 2D mutant or phosphorylated Cdc6 WT incubated with Plk4.

Comment 15. Figure 6 e. The blot of the levels of PLK4 shows little difference in PLK4 levels and thus the interpretation of this experiment is not justified.

Answer 15: We have repeated the immunoprecipitation between Cdc6 and Sas-6 after Plk4 depletion with a different Plk4 siRNA (new Supplementary Fig. 6f) which is more efficient than the Plk4 siRNA (original Supplementary Fig. 5c) we used in the previous manuscript. The result showed that the interaction between Cdc6 and Sas6 is significantly decreased upon Plk4 depletion.

Comment 16. Page 13. The is inappropriate conclusion here. Just because knock down of SAS-6 blocks centriole duplication in the absence of CDC6 does not mean, one way or the other, that CDC6 directly blocks SAS-6. It means that SAS-6 is required for centriole duplication. The same might be true for HU induced amplification and one would not conclude that SAS-6 is regulated by HU!

Answer 16: We thank this reviewer for pointing this out. We have now tested whether Cdc6 directly blocks Sas6 activity by in vitro protein-protein interaction assay. First, we tested and found that Cdc6 binds to Sas-6 protein directly in vitro (new Fig. 6b). Second, we tested in vitro whether Cdc6 phosphorylation regulates the direct binding between Cdc6 and Sas-6 (new Fig. 6h). The result showed that Cdc6 phosphorylation either by incubated with Plk4 or Cdc6 2D mutant inhibits the interaction between Cdc6 and Sas-6. Third, we further tested whether Cdc6 phosphorylation regulates Sas-6 interaction with STIL (new Fig. 6i). STIL was reported to interact with Sas-6 in a Plk4 phosphorylation-dependent manner on procentriole formation site to induce centrosome duplication (J Cell Biol. 2015 Jun 22;209(6):863-78; Nat Commun. 2014 Oct 24;5:5267.). The result showed that Cdc6 phosphorylation either by incubated with Plk4 or Cdc6 2D mutant facilitates a strong interaction between Sas-6 and STIL, while the unphosphorylatable Cdc6 2A mutant inhibited the interaction between Sas-6 and STIL. Therefore, we concluded that Cdc6 directly inhibits Sas-6 from forming a stable complex with another procentriole formation core protein STIL, while Cdc6 phosphorylation by Plk4 suppresses the inhibition of Cdc6 on Sas6-STIL complex formation and activates centrosome duplication.

Comment 17. Page 14 and abstract. The authors conclude that CDC6 binds SAS-6, which is fine, but they also conclude that it inhibits SAS-6 activity. Can they point to the data that

supports this conclusion? (note point above). The only data that is relevant here is Figure 6 f, but I note that overexpression of Myc-SAS-6 in the top panel yields a large SAS-6 cluster, and indeed CDC6 blocks this, but the CDC6-2D mutant does not yield this large cluster, only extra centriole pairs. This is not the same as phosphorylation of CDC6 by PLK4 regulating SAS-6.

Answer 17: We have now confirmed that Cdc6 directly binds to Sas-6 by in vitro protein-protein interaction (new Fig. 6b). We also proved in vitro that Cdc6 phosphorylation negatively regulates the direct binding between Cdc6 and Sas-6 (new Fig. 6h). We further found that Cdc6 directly inhibits Sas-6 from forming a stable complex with another procentriole formation core protein STIL, while Cdc6 phosphorylation by Plk4 suppresses the inhibition of Cdc6 on Sas6-STIL complex formation and activates centrosome duplication. (new Fig. 6i). These results are also stated in the above point.

We have repeated the experiment of centriole overduplication in cells co-expressing Myc-Sas-6 and GFP-Cdc6 WT, 2A, or 2D mutant (new Fig. 6d,e). Both GFP vector transfected and GFP-Cdc6 2D mutant transfected cells showed overduplicated centriole clusters indicated by centrin1.

Reviewers' Comments:

Reviewer #1 (Remarks to the Author)

I have read the newest version of the manuscript by Xu et al and they have sufficiently addressed my concerns. I do suggest the following as additional improvement to the manuscript:

1. The newly presented data in figure 6i is very exciting. I would like the author to clarify if STIL is also phosphorylated in their experiment. If STIL is phosphorylated, then this suggests that STIL phosphorylation by Plk4 is not sufficient for Sas6 interaction (His-A2 result). If STIL is not phosphorylated in their assay, then it suggests that STIL-Sas6 interaction does not require PLK4 phosphorylation (His-2D result). Either way, these results significantly advance our understanding of the STIL-Sas6 interaction and must be discussed.
2. All the images showing blow-ups of centrioles/centrosomes in insets are artificially and unacceptably contrasted. In the lower magnification image, background fluorescence is very clear, yet the in the blow-up, background is made black. The images should not be enhanced in this way to allow the reader to properly assess localization, especially in the cases where they claim no protein is detected.

Reviewer #2 (Remarks to the Author)

The revised manuscript from Xu et al. is much improved and the authors have successfully addressed my major concerns. I therefore strongly support publication in Nature Communications, although I have a number of minor points that the authors might like to consider prior to publication.

1. The authors description of the centriole duplication in the Introduction is slightly inaccurate. They describe a conserved pathway of duplication in worms, flies and humans and state that SPD-2/Cep192 is the crucial recruiter of Plk4. It is clear that flies do not use SPD-2, but instead use Asl (Cep152 in humans) (e.g. Novak et al., *Curr. Biol.*, 2010), while human cells can use both Cep192 and Cep152 (Kim et al., *PNAS*, 2013; Sonnen et al., *JCS*, 2013).
2. I think it is important that the authors give some estimate of the resolution they are achieving in their 3D-SIM and STED experiments. This is important as the authors claim co-localisation, but it is not clear that the dots they are seeing are not actually below the resolution of their microscope systems.
3. The authors claim the S30 and T527 residues show the highest homology to the Plk4 consensus site, but I couldn't see any information on what they are using to predict Plk4 phosphorylation sites.
4. Lines 321-325 there seems to be some confusion in text about whether GFP and/or mCherry tagged proteins are being used.
5. The experiments in which Cdc6 is co-depleted with either Plk4 or Cdc6 need to be documented better. In both cases blots should be shown to confirm the co-depletion, and that the co-depletion of Cdc6 is not simply making the depletion of Plk4 or Sas-6 less efficient. The authors should also quantify the effect of just depleting Sas-6 (they do show this for the Plk4 experiments), as they present this data as though the depletion of Sas-6 is specifically eliminating the Cdc6-depletion-induced centriole amplification, when of course Sas-6 depletion would be expected to inhibit all forms of centriole duplication. The experiments that follow nicely show how Cdc6 might influence

Sas-6, but the description of this first experiment should be toned down.

Reviewer #3 (Remarks to the Author)

In the response Reviewer 1 comment and Reviewer 3 comment 14 regarding the direct interaction between Cdc6 and Sas-6, the authors present a co-immuno-precipitation experiment with in vitro transcribed (not purified) proteins (new figure 6B). An important control is missing in this experiment, namely showing that the anti-Cdc6 antibody does not cross react with SAS-6 protein directly. A normal experiment would include a lane in which Cdc6 is left out and showing the antibody does not co-precipitate Sas-6. This should be done as it is standard for biochemistry.

It is noted that the anti-Cdc6 immuno-precipitation in Figure 6h shows varied levels of Sas-6 co-precipitated depending on Cdc6 phosphorylation status, arguing against the cross-reactivity. Rather than hold up the paper further, the authors could mention this in the text.

Point-by-point answers

Reviewers' comments:

Reviewer #1 (Remarks to the Author):

I have read the newest version of the manuscript by Xu et al and they have sufficiently addressed my concerns. I do suggest the following as additional improvement to the manuscript:

Comment 1) The newly presented data in figure 6i is very exciting. I would like the author to clarify if STIL is also phosphorylated in their experiment. If STIL is phosphorylated, then this suggests that STIL phosphorylation by Plk4 is not sufficient for Sas6 interaction (His-A2 result). If STIL is not phosphorylated in their assay, then it suggests that STIL-Sas6 interaction does not require PLK4 phosphorylation (His-2D result). Either way, these results significantly advance our understanding of the STIL-Sas6 interaction and must be discussed.

Answer 1: We thank this reviewer's good suggestion. We have now included the detailed methods for this experiment in original Fig. 6i (new Fig. 6j). Briefly, we first incubated the indicated protein with or without Plk4 protein in kinase buffer with ATP to allow kinase reaction and then performed the following immunoprecipitation. To test whether STIL is phosphorylated by Plk4 in this experiment, we detected the phosphorylation status of STIL by phosphoserine antibody after immunoprecipitation of STIL from the reaction (new Supplementary Fig. 6i). The result showed that STIL is phosphorylated by Plk4 kinase (new Supplementary Fig. 6i). This result suggests that phosphorylation of STIL was not sufficient for its interaction with Sas-6, as unphosphorylated Cdc6 2A mutant inhibited the interaction between STIL and Sas-6, indicating Cdc6 is involved in the regulation of STIL-Sas-6 interaction.

Comment 2) All the images showing blow-ups of centrioles/centrosomes in insets are artificially and unacceptably contrasted. In the lower magnification image, background fluorescence is very clear, yet the in the blow-up, background is make black. The images should not be enhanced in this way to allow the reader to properly assess localization, especially in the cases where they claim no protein is detected.

Answer 2: We thank this reviewer for pointing out this inappropriate adjustment. We have now replaced all the blow-up images (Fig. 1b,f; Fig. 2c,e; Fig. 3a; Fig. 5a,d; new Fig. 6e; Supplementary Fig. 1a,b,h,i; Supplementary Fig. 3c; new Supplementary Fig. 5a and new Supplementary Fig. 6c.) with the original contrast without any artificially contrast adjustment.

Reviewer #2 (Remarks to the Author):

The revised manuscript from Xu et al. is much improved and the authors have successfully addressed my major concerns. I therefore strongly support publication in Nature Communications, although I have a number of minor points that the authors might like to consider prior to publication.

Comment 1) The authors description of the centriole duplication in the Introduction is slightly inaccurate. They describe a conserved pathway of duplication in worms, flies and humans and state that SPD-2/Cep192 is the crucial recruiter of Plk4. It is clear that flies do not use SPD-2, but instead use Asl (Cep152 in humans) (e.g. Novak et al., Curr. Biol., 2010), while human cells can use both Cep192 and Cep152 (Kim et al., PNAS, 2013; Sonnen et al., JCS, 2013).

Answer 1: Thank you for pointing out the inaccuracy. We are sorry about the misleading description about the conserved pathway of centrosome duplication. We have now replace the original description with “SPD-2 (Cep192 in human) is required to recruit ZYG-1(Plk4 in human) in *C. elegans*, Asl (Cep 152 in human) is required to recruit Sak (Plk4 in human) in *Drosophila*, and Cep192 and Cep152 are both required to recruit Plk4 in human.”.

Comment 2) I think it is important that the authors give some estimate of the resolution they are achieving in their 3D-SIM and STED experiments. This is important as the authors claim co-localisation, but it is not clear that the dots they are seeing are not actually below the resolution of their microscope systems.

Answer 2: This is a good point. We have now added the resolutions of the 3D-SIM and STED images in the methods. The 3D-SIM system that we used has a 120 nm highest resolution, and in our 3D-SIM images, the resolution was 130 ± 3 nm measured following a Gaussian distribution with a full width at half maximum (FWHM). The STED system that we used has a 50 nm highest resolution. In our STED images, the resolution was 90 ± 5 nm measured following a Gaussian distribution with a full width at half maximum (FWHM). The centriole dots shown in these 3D-SIM and STED images are approximately same to the resolution of our microscope systems and co-localization can be concluded.

Comment 3) The authors claim the S30 and T527 residues show the highest homology to the Plk4 consensus site, but I couldn't see any information on what they are using to predict Plk4 phosphorylation sites.

Answer 3: We have now added the method we used to predict this two sites in the results. We used GPS 3.0 (Group-based Prediction System, version 3.0), a kinase-specific phosphorylation site prediction software, for Plk4 phosphorylation sites prediction. S30 and T527 are the two sites with highest scores (Mol Cell Proteomics. 2008 Sep;7(9):1598-608.).

Comment 4) Lines 321-325 there seems to be some confusion in text about whether GFP and/or mCherry tagged proteins are being used.

Answer 4: We are sorry about this confusion. In Fig. 5d,e, we co-overexpressed mCherry tagged Cdc6 or Cdc6 mutants with GFP tagged Plk4, but we made a mistake on the

description. We have now changed “Compared to 52.7% of the control cells co-expressing **GFP** and Plk4 showing the centriole amplification, only 26.6% of the cells co-expressing **GFP-Cdc6 WT** and Plk4 exhibited the centriole amplification” to “Compared to 52.7% of the control cells co-expressing **mCherry** and Plk4 showing the centriole amplification, only 26.6% of the cells co-expressing **mCherry-Cdc6 WT** and Plk4 exhibited the centriole amplification”

Comment 5) The experiments in which Cdc6 is co-depleted with either Plk4 or Cdc6 need to be documented better. In both cases blots should be shown to confirm the co-depletion, and that the co-depletion of Cdc6 is not simply making the depletion of Plk4 or Sas-6 less efficient. The authors should also quantify the effect of just depleting Sas-6 (they do show this for the Plk4 experiments), as they present this data as though the depletion of Sas-6 is specifically eliminating the Cdc6-depletion-induced centriole amplification, when of course Sas-6 depletion would be expected to inhibit all forms of centriole duplication. The experiments that follow nicely show how Cdc6 might influence Sas-6, but the description of this first experiment should be toned down.

Answer 5: We thank this reviewer for all these suggestions. We have now included the depletion efficiency of Plk4 single depletion or Cdc6 co-depletion with Plk4 by western blotting in new Supplementary Fig. 5e. Cdc6 and Sas6 single depletion and co-depletion efficiency is shown in new Supplementary Fig. 6e. These results confirmed that co-depletion of Cdc6 does not influence the depletion efficiency of Plk4 or Sas-6.

We also quantified the effect of Sas-6 and Cdc6 single depletion or co-depletion by counting the centriole number revealed by centrin1 (new Fig. 6c,d). The results showed that co-depletion of Sas-6 and Cdc6 eliminated the Cdc6 depletion-induced centriole amplification, and even resulted in large percentage of cells with single centriole similar to Sas-6 depletion (new Fig. 6c,d). Since Sas-6 depletion inhibits all forms of centriole duplication, we have now changed our conclusion of this experiment from “Cdc6 might prevent centriole amplification by inhibiting Sas-6” to “These results suggested that Sas-6, as an essential centrosome duplication initiator, its depletion overrides the Cdc6 depletion-induced centriole amplification”.

Reviewer #3 (Remarks to the Author):

In the response Reviewer 1 comment and Reviewer 3 comment 14 regarding the direct interaction between Cdc6 and Sas-6, the authors present a co-immuno-precipitation experiment with in vitro transcribed (not purified) proteins (new figure 6B). An important control is missing in this experiment, namely showing that the anti-Cdc6 antibody does not cross react with SAS-6 protein directly. A normal experiment would include a lane in which Cdc6 is left out and showing the antibody does not co-precipitate Sas-6. This should be done as it is standard for biochemistry.

It is noted that the anti-Cdc6 immuno-precipitation in Figure 6h shows varied levels of Sas-6 co-precipitated depending on Cdc6 phosphorylation status, arguing against the cross-reactivity. Rather than hold up the paper further, the authors could mention this in the text.

Answer: We thank this reviewer for this good suggestion. We have now included this

important control in new Fig. 6b. The lane in which Cdc6 is left out showed that Cdc6 antibody did not co-precipitate Sas-6, indicating Cdc6 antibody does not react with Sas-6 protein. We have also added the relevant description about this negative control in the text. Accordingly, in the text describing new Fig. 6i (original Fig. 6h), we also added that “Notably, the varied levels of co-precipitated Sas-6 depending on Cdc6 phosphorylation status excluded the possibility of cross-reactivity between Cdc6 antibody and Sas-6 protein.”

REVIEWERS' COMMENTS:

Reviewer #1 (Remarks to the Author):

I support publication.

--

Reviewer #2 (Remarks to the Author):

The re-revised paper is fine. I did not really need to see it again.

--

Reviewer #3 (Remarks to the Author):

I have reviewed the response letter and the new data and I am OK with this paper moving forward.